# The Characteristics, Distribution, Function, and Origin of Alternative Lateral Horse Gaits

**DOI:** 10.3390/ani13162557

**Published:** 2023-08-08

**Authors:** Alan Vincelette

**Affiliations:** Department of Pretheology, St. John’s Seminary, 5012 Seminary Road, Camarillo, CA 93021, USA; avincelette@stjohnsem.edu

**Keywords:** horse gaits, lateral gaits, locomotive kinematics, locomotive biomechanics, horse evolution, gaited horse breeds

## Abstract

**Simple Summary:**

Horse breeds with alternative lateral gaits, such as the running walk, rack, broken pace, hard pace, and broken trot, were important historically and are popular today among equestrians for their trail or pleasure gait and at horse shows. This article reviews what is known about these gaits, including their origin, distribution, kinematics, functional, and biomechanical advantages. It incorporates evidence from art, human history, fossil equid trackways, and genetics to provide a comprehensive overview of our current state of knowledge about the evolution and development of alternative lateral gaits as well as variations in their expression.

**Abstract:**

This article traces the characteristics, origin, distribution, and function of alternative lateral horse gaits, i.e., intermediate speed lateral-sequence gaits. Such alternative lateral gaits (running walk, rack, broken pace, hard pace, and broken trot) are prized by equestrians today for their comfort and have been found in select horse breeds for hundreds of years and even exhibited in fossil equid trackways. After exploring the evolution and development of alternative lateral gaits via fossil equid trackways, human art, and historical writings, the functional and genetic factors that led to the genesis of these gaits are discussed. Such gaited breeds were particularly favored and spread by the Scythians, Celts, Turks, and Spaniards. Fast and low-swinging hard pacing gaits are common in several horse breeds of mountainous areas of East and North Asia; high-stepping rack and running walk gaits are often displayed in European and North and South American breeds; the broken pace is found in breeds of Central Asia, Southeast Asia, West Asia, Western North America, and Brazil in South America; and the broken trot occurs in breeds of North Asia, South Asia, the Southern United States, and Brazil in South America, inhabiting desert or marshy areas.

## 1. Introduction

Members of the genus *Equus*—including domesticated caballine horse breeds (*Equus ferus caballus*), the likely non-domesticated Przewalski Horse (*Equus ferus przewalskii*), zebras, and African and Asian wild asses—standardly perform the three so-called “natural” gaits of walk, trot, and canter/gallop [1,2,3,4].

The walk is the standard slow gait of the horse occurring at around 1.4–1.8 m/s that involves independent movement of each limb in a lateral footfall sequence: left hind, left front, right hind, right front. It is often called a “stepping”, “square”, or “singlefoot” gait as it possesses a lateral advanced placement (or limb phasing value) close to 0.25 with the limbs sequentially setting down in near quarter intervals of the stride duration. The walk yields four audible even beats with the movements of the right legs mirroring those of the left legs. Hence, the walk can be categorized as a symmetrical, even four-beat lateral sequence “singlefoot” gait. It has a duty factor of 0.70–0.60 (proceeding from slower to faster speeds)—indicating that the hind limbs make contract with the ground during 60–70% of the stride duration—and the walk has bipedal support, alternating between diagonal and ipsilateral or tripedal support at all times [5,6,7,8].

The trot (or jog if performed in slow manner) is the standard intermediate speed gait of the horse. It occurs at speeds of ca. 2.5–6.5 m/s, up to 8.5–12.0 in harness racing. It is a symmetrical diagonal sequence and diagonal couplet gait: left hind + right front, right hind + left front. Though it yields two even beats, at slower speeds, the front hoof often lands just before the hind diagonal hoof, and at higher speeds, the hind hoof lands just before the front diagonal hoof but not enough to make an audible difference. It has a limb phasing value, or lateral advanced placement, of 0.50 (as limb pairs land at around half the duration of the stride) and a duty factor or hind leg support phase of 0.55–0.30 (from slower to faster speeds), resulting in many diagonal bipedal support phases, some unipedal support phases, and periods of suspension when all four feet are off the ground at the same time [5,7,8,9,10,11,12]. The trot often occurs with elevated and animated front legs as with the park trot of the Morgan, the Spanish ‘walk’ of the Andalusian, or the collected passage in dressage.

The natural fast gait of the horse is an asymmetrical gait involving the coordination of contralateral limbs and suspended phases where all four feet are off the ground, which occurs around 9.0–15.0 m/s, up to 17.0–19.0 m/s in horse racing, where it is called the run or Renngalopp in German. This gait is variously called the canter or gallop, but here the two terms will be distinguished. If performed at the lower to middle portions of the velocity range, the gait often takes the form of a three-beat canter wherein a hind limb contacts the ground, followed by a diagonal limb pair, and ending with contact of the front leading limb, i.e., left hind, right hind + left front, right front in a right-lead canter. The canter thus has periods of unipedal and diagonal bipedal limb support and can occasionally even have periods of tripedal support (i.e., in a slower lope) [5,13]. When performed at the faster end of the range, the gait often transitions to an uneven four-beat gallop, though in a very fast run or racing speed, the two contralateral hoof strikes can occur so close together that the gait almost sounds like it only has two beats. In equines, the gallop takes the form of a transverse gallop involving repeated strikes of mirror-image contralateral couplets, i.e., left hind, right hind, left front, right front in a right-lead gallop or the reverse in a left-lead gallop. The gallop has periods of unipedal and contralateral bipedal support as well as phases where all four limbs are off the ground together. The duty factor, or hind limb support phase as percentage of stride duration, varies from 0.30 in the canter to 0.20 in a fast gallop [14,15].

In addition to these “natural” gaits, several “gaited” horse breeds display additional intermediate speed symmetrical and lateral-sequence gaits (the running walk, rack, broken pace, hard pace, and broken trot) [16,17,18,19], or what I shall call alternative lateral gaits. The broken pace (stepping pace or amble) and broken trot (fox trot) involve four uneven beats, as the even and nearly simultaneous landing of ipsilateral or diagonal couplets occurring in the hard pace and trot are uncoupled. The running walk and rack (tölt) are even four-beat or “singlefoot” gaits, with the former being a hyperextended walk relying on inverted-pendulum swinging mechanics and the latter a bouncing gait relying upon spring mass mechanics of the ligament system. In contrast with the canter/gallop, trot, and hard pace, the running walk, rack, broken pace, and broken trot maintain ground contact at all times, with one to three, or even all four, hooves, despite being intermediate speed gaits. This makes them quite comfortable for the rider and useful in maintaining balance when traversing uneven terrain. This article reviews what is known about such alternative lateral gaits, including their characteristics, origin, history, and genetics.

## 2. Alternative Lateral Gaits of the Horse

In addition to the “natural” gaits of walk, trot, and canter/gallop, there are additional gaits (sometimes called “artificial” or supplemental) spontaneously displayed in certain domestic horse breeds (i.e., gaited horses) [20,21,22,23]. All of these additional gaits are symmetrical lateral sequence gaits, i.e., possessing a footfall pattern of left hind, left front, right hind, right front. Because these gaits are found in certain fossil equids, are often performed in the foals of certain modern horse breeds without any training, and sometimes replace other gaits, such as the trot or gallop, I shall here label them alternative lateral gaits rather than “artificial” or supplemental.

Five types of alternative lateral horse gaits can be distinguished, with certain variants therein, the running walk (or flat walk at slower speeds), rack (often called the tölt in Europe), the broken pace (or stepping pace), the hard pace, and the broken trot (often called the fox trot). These gaits are distinguishable in terms of audible beats (number and chronicity), temporal kinematic parameters (advanced placement and lift-off of consecutive lateral or diagonal limbs as a percentage of total stride duration, whether temporally the closest couplets are diagonal or ipsilateral, and duty factor or hind limb stance duration), support structures (number of hooves on the ground at the same time in stance phases), linear parameters (footprint patterns, stride or “cycle” length, and track gauge width), joint angles, body movements (in particular head and croup), and biomechanics (inverted pendulum vs. spring mass vs. collisional).

### 2.1. The Running Walk

The running walk is an accelerated outstretched walk wherein the biomechanics and limb sequence of the standard walk are largely retained, but the hind limbs extend forward under the horse’s body before pushing off in order to propel the horse in an intermediate-speed and overstepping gait. Like the walk it is a four-beat lateral sequence gait, and preferably square or “singlefoot” (especially in the show arena) with four even beats, though occasionally a lateral couplet with ipsilateral hooves landing closer together in time than diagonal hooves. Again, the running walk, like the walk, biomechanically involves swinging the horse’s legs in an inverted pendulum-like manner, generation of forward motion through muscular effort in pushing limbs off the ground, and minor flexing or extending of the leg joints. However, the bulk of the forward propulsion of the running walk comes from the hind limbs extending forward under the horse’s body and overstepping the landing point of the ipsilateral front hooves. In order to maintain balance, and not waste kinetic energy, horses in the running walk will nod their head up and down, often causing the ears to flop up and down as well. Thus, a horse displaying a running walk will have a low outstretched stride in the rear limbs but an elevated stride in the front limbs with knee joints often quite flexed and the front hooves uplifted. There is some debate as to whether the “running walk” is properly spoken of as a walk or a run; however, the present author finds no solid reason to discard the traditional name of running walk—especially as its speed can attain 2.5–4.0 m/s, at which point most non-gaited horses will have transitioned to a trotting gait, yet involves inverted pendulum biomechanics [24,25,26].

The two forms of the running walk are the slower flat walk and the faster running walk proper, i.e., the hard running walk. The flat walk (or sometimes called the flat-foot or flat-footed walk) takes place at around 1.8 to 2.2 m/s with a stride duration of 1.10–0.95 s and a stride length of 1.8 to 2.4 m. The flat walk is often slightly uneven (the ratio of ipsilateral over diagonal step time is 0.60–0.70) and so involves lateral couplets, though as much of an even singlefoot gait as possible is encouraged and an overly pacy walk or “camel walk” discouraged in the show arena. In the flat walk, there is a period during which a front or hind hoof lies flat on the ground (hence the name of the gait), and there is some head nod (vertical head displacement of 11–19 cm) and a little forward-to-back movement in the saddle but minimal up-and-down movement (croup vertical displacement around 8 cm). The flat walk contains many tripedal support phases (ca. 0.40–0.70 of the stride length) with alternating lateral and diagonal bipedal periods of support [27,28,29].

The running walk proper or hard running walk is the faster version of the gait, taking place around 2.2 to 4.0 m/s, with a stride duration of 0.85–0.65 s, and a stride length of 2.1 to 2.9 m. The hard running walk is ideally quite even (ratio of ipsilateral over diagonal step times of 0.70–0.90), but the front limbs break over and lift off the ground just before the “heels” of the hind hooves land, and so it is not flat footed. Hence, while the running walk has a lateral advanced placement close to 0.25 (0.23 to 0.17 at slower to faster speeds), it possesses a fairly low lateral advanced lift-off (0.10–0.19) and a fairly high diagonal advanced lift-off (0.31 to 0.48). There is also substantial head nodding that occurs with the running walk, which helps the horse maintain balance (vertical head displacement 10–24 cm). The back of the horse is fairly level (i.e., unrounded) in the running walk with very little up-and-down movement of the rider in the saddle (croup vertical displacement ca. 3–6 cm), and only a slight front to back rocking motion in the saddle. The running walk has minimal tripedal support phases (0.15 to 0.05 at slower to faster speeds) compared to other lateral gaits and may have occasional periods of unipedal support (0.02–0.08) at very fast speeds, with two front feet and one hind foot in the air simultaneously. The running walk tends to leave a trackway of four isolated prints without obvious ipsilateral or diagonal pairs [23,30,31,32,33].

The running walk is a very comfortable gait for the rider and prized in horse breeds where a long duration intermediate-speed gait is desired. It is also a very sure-footed gait with periods of tripedal support at slower speeds and with one limb always on the ground. Unfortunately, the biomechanics of the running walk including energetic consumption and ground contact forces remain to be studied; however, Holt [29] found that the average respiratory rate of Tennessee Walking Horses after a flat walk at 1.7 m/s was considerably less (26–27 breaths/minute) than that of a trot at 3.3 m/s (75–80 breaths/minute) or trot at 5.0 m/s (95–100 breathes/minute) [34]. Hence, it may well be the case that the running walk, with less vertical fluctuation of its center of mass, is more energy efficient than the trot at intermediate speeds and indeed than other alternative lateral gaits.

The running walk gait occurs in many breeds of the Southern United States, but most famously in the Tennessee Walking Horse [30,31,32,33]. In the Tennessee Walking Horse, the running walk occurs with lots of head nodding, either with slight elevation and animation of the front limbs in a pleasure or road gait or with great elevation and animation of the front limbs in a show gait. Indeed, at its extreme, a very high-stepping running walk, or “big lick”, is induced through the use of modified shoes and, in the past, the censured practice of soring of the horses’ feet. The Tennessee Walking Horse is also capable of a fast running walk including periods of unipedal support. The running walk also occurs in the American Walking Pony though with less head nodding (where it is called a pleasure walk when slower and a merry walk when faster), Florida Cracker, McCurdy Plantation, Smokey Valley (where it is labeled the soft prance), and Tiger Horse, as well as in genetic offshoots of the Tennessee Walker, including the North American Curly, Tennuvian (cross of Tennessee Walker with Peruvian Paso), Utah Walkony (cross of Tennessee Walker with pony), and Walkaloosa (cross of Tennessee Walker with Appaloosa) [35,36,37]. A running walk is occasionally found in other gaited breeds of the Southern United States, including the Banker, Kentucky Mountain Saddle, Morgan, Rocky Mountain, Spanish Mustang, Spanish Colonial, and Spotted Saddle Horse, though typically with little elevation and animation of the front limbs. The running walk gait, in addition, has also made its way up north where it is sometimes found in the Canadian Pony of the Americas, Montana Travler, and Sable Island Horse (where it is called the prance), as well as south into Mexico with the Sierra Tarahumara. The Galiceño of Mexico is also said to perform the running walk, but one study of the breed only found a standard walking gait at ca. 1.36 m/s [38], while other reports and illustrations seem to indicate it possesses more of a rack or broken pace [20]. The expression “running walk” is also occasionally used for the alternative lateral gait common in Greek and Turkish breeds, but their gait is more properly labeled a broken pace.

The running walk takes a unique form in South America and the Spanish Caribbean. In South America, the running walk occurs in the Andean and Peruvian Paso Horse where it is called the paso llano (a shortening of paso castellano) or the andadura rota. It occurs, however, in a unique form having less head nod and hind leg extension and with the forelegs moving in an outward sweeping arc, or what is called termino, with some elevation and sharply angled knee joints (agudez) and animation (brio). The paso llano tends to be performed more quickly than the standard running walk with a stride duration of 0.63–0.57 (slower to faster speeds) and with a slightly shorter stride length of ca. 1.7–1.9 m, and occasionally has slightly diagonal couplets (lateral advanced placement of 0.27–0.28) [30,35,36,39]. The Peruvians are very precise in categorizing different forms the running walk can take, namely, golpeado or slow moving with quick understeps, little forward progress, slight unevenness, diagonal couplets, more diagonal bipedal support than lateral bipedal support phases, and discomfort for the rider; picado or even with some speed and capped prints; and gateado or fast with overstep, slight unevenness with lateral couplets, longer hind limb support phases than front limb support phases, more lateral bipedal support than diagonal bipedal support phases, and great comfort for the rider. A similar running walk with termino was found in the Abaco Barb of the Bahamas and likely the extinct Spanish Jennet.

### 2.2. The Rack

The rack is a lateral sequence four-beat gait which varies from having lateral couplets to being square or “singlefoot”, the latter being prized in the show arena. The rack has a shorter hind step than the running walk with less extension and is usually performed at a quicker rate (stride duration 0.70–0.42 s with increasing velocity) and faster speed (2.5–6.0 m/s, up to 10.6 in racing horses) with a longer stride length (2.0 to 3.6 m, up to 4.5 to 6.0 m in racing horses). The rack has a lateral advanced placement of around 0.24–0.18 and a diagonal advanced placement of 0.34–0.24 (from slower to faster speeds). It is typically an elegant gait in which the horse holds its head high with little nod, and often its forelegs are quite elevated and display much animation. The rack has occasional tripedal support at slower speeds (i.e., a saddle rack) with understepping or capping of ipsilateral hoof tracks, but at faster speeds, in a hard rack or speed rack, there are only bipedal (mostly ipsilateral with some diagonal) and unipedal support phases with overstepping of ipsilateral limb tracks to such an extent that diagonal pairs of hoof impressions are close together in trackways. The rack is also a narrow-gauge gait wherein the legs move inward closer to the center of the horse’s body so the interior straddle is often negative in value as hind limb impressions cross the centerline parallel to the direction of travel [30,40,41,42,43,44,45].

Like the running walk, the rack, with periods of tripedal support when slow and with one leg on the ground at all times even when fast, seems well adapted for slippery, rugged, uneven, or steep terrain. Hence the rack tends to be associated with breeds inhabiting mountainous or hilly regions. It is also fairly comfortable for the rider. The back of the horse tends to be ventroflexed (concave) in the rack, and while the rider does experience some up-and-down motion in the saddle (croup vertical displacement of 3.0 to 6.5 cm), as evidenced by the shaking of the horse’s tail, there is not nearly as much up and down motion as in the trot (croup vertical displacement of 5.0–13.0 cm), as most of the up-and-down motion is absorbed at the back of the horse, though there can be some slight swaying from side to side.

Biomechanically, the racking gait utilizes a spring mass mechanism involving release of energy of stretched ligaments to help propel the horse forward; however, there is always at least one limb that maintains ground contact, and often two [46,47,48]. For this reason, peak ground contact forces (especially for the hind limbs) are higher in the trot than in the tölt (rack) at speeds of ca. 3.0–4.0 m/s (9.4–10.0 N/kg for front limbs and 6.6–7.0 N/kg for hind limbs in the tölt, and 8.9–11.7 N/kg for front limbs and 7.5–10.0 N/kg for hind limbs in the trot) [49,50,51]. Hence, the tölt allows for high speeds at short distances without stressing the leg joints as much as the trot or hard pace. The rack, however, is not as efficient a gait as trot, hard pace, gallop, or even the broken trot or broken pace, for intermediate-speed long distance travel. The energetic cost of the rack (or tölt), in fact, is 4.8–5.5% greater than that of the trot at high velocities and around 15% greater than that of the hard pace or gallop [49,50]. Stefánsdóttir [52] found that while mean heart rates were similar for Icelandic Horses in the tölt (132, 153, 180 BPM) and trot (131, 154, 186 BPM) at speeds of 3.2, 4.1, and 5.5 m/s, mean lactate blood concentrations were higher in the tölt (1.07, 1.48, 4.66 mmol/L) than in the trot (0.92, 1.27, 4.92 mmol/L) at speeds of 3.2 and 4.1 m/s but not at 5.5 m/s. Respiratory rate, however, though slightly lower in the tölt than the trot immediately after locomoting from 3.2 to 5.5 m/s (70 vs. 75 breaths/minute), was higher 30 min later (29 vs. 25 breaths/minute). They argue that this yields a larger recovery time for the tölt than trot.

Though not found in most breeds of southern Europe, the rack is often found in the Portuguese Garrano Horse (where it is called the passo travado and takes the form of a hard rack with periods of unipedal support) and sometimes in the Alter Real of Portugal.

The racking gait is well known and best studied in the Icelandic Horse (where it is known as the tölt) and can be performed either as a saddle rack with periods of tripedal support or as a hard rack with unipedal support phases, though the front legs are not as elevated as in the American Saddlebred [40,41,42,43,44,45]. The rack (or more properly tölt) is also found in genetic offshoots of the Icelandic Horse including the Danish Faeroe Island Pony and the German Aegidienberger (cross of Icelandic with Peruvian Paso).

The rack is also a prominent gait in many horse breeds of the Southern United States. The racking gait is prized in some individuals of the American Saddlebred breed, where it tends to be more trained than natural and takes the form of either a saddle rack with very elevated and animated forelimbs and tripedal and bipedal support structures or a hard rack with only bipedal and unipedal support structures and is usually a trained gait [35,36,37,53,54]. The rack also occurs in the Florida Cracker (i.e., the coon rack), where it is a quick-stepping saddle rack with extensive lateral bipedal support phases due to a large diagonal advanced lift-off (ca. 0.42); Kentucky Mountain Saddle Horse (i.e., a mountain pleasure rack), which has a saddle rack with front legs moving with low elevation and less animation; Kentucky Natural Gaited Horse; McCurdy Plantation Horse (the McCurdy lick, plantation gait, or saddle gait), which often takes the form of a hard rack with some but not extreme front limb elevation; Mountain Pleasure Horse, with a saddlerack (trail or mountain pleasure rack with some but not extreme front leg elevation or head left); National Show Horse (cross of Arabian with Saddlebred), which has a rack similar to the Saddlebred with elevated front limbs and animation; National Spotted Saddle Horse; North American Singlefooting Horse, which can perform a slower saddle rack with some elevation of front legs (trail or country rack), but is most known for its hard or speed rack occurring at very fast speeds with extensive unipedal support phases (road gait); Racking Horse, which also has a hard rack or speed rack (the style rack) with extensive unipedal support phases as well as a large diagonal advanced lift-off of ca. 0.40 and longer hind than front limb support phases; Rocky Mountain Horse (mountain rack), which can take the form of a slower saddle rack (show gait) or a faster hard rack (pleasure gait), though both are performed with relatively low elevation and animation of the front legs; Smoky Valley Horse (traveling gait), which can be either a saddle rack or hard rack; and the Virginia Highlander, which usually performs a hard or speed rack [18,23,30,35,36].

The rack is also sometimes found in the American Walking Pony, Morgan, Newfoundland Pony and Sable Island Horse of Canada, Spanish Mustang and Spanish Colonial Horse (with less elevated front legs), Sierra Tarahumara of Mexico, Tennessee Walking Horse (often with elevated front legs), and the Walkaloosa. The rack may be seen on occasion in the Azteca of Mexico, the Banker (also called the Corolla) of the United States, and probably in the Galiceño of Mexico [20].

South American horses and their offshoots, including the American Paso Fino, Colombian Paso Fino, Colombian Trocha Pura, Puerto Rican Criollo, and Puerto Rican Paso Fino, are named after their paso fino racking gait (classic fino, fino clásico, or show gait). The classic fino is typically a very collected rack performed with quick low steps, little forward momentum (i.e., a very slow speed), and tripedal support phases. Accordingly, such gaits typically have a short stride length (ca. 0.6 to 1.1 m) and stride duration (0.39–0.35 s as velocity increases) [30,35,36,55,56,57]. The Paso Fino breeds can also perform faster versions of the rack called the paso corto with moderate speed and extension and the paso largo with great speed and extension, though these gaits most often take the form of a broken trot. The Cuban Paso performs a similar rack to the classic fino, where it is called the marcha fina y gualdrapeada or paso del gualdrapeo. Paso Fino horses also often perform what is called a “flat walk”, but this is really just a standard walk rather than a slower variant of the highly extended running walk.

Several Brazilian horse breeds perform a broken pace (marcha picada), or broken trot (marcha batida), but recently a square rack (the marcha de centro, marcha de intermediária) has become popular in such breeds as the Campeiro, Campolina, Mangalarga Marchador, Mangalarga Paulista, and Mangolina. A rack can also occur in the Venezualan Criollo (also called the Llanero).

The rack is also a common gait in South African horses. Not only is it found in Saddlebreds imported from the United States but also in more native breeds, including the Basuto, Cape Boer, and Nooitgedacht, which often display a saddle rack termed the trippel in light of its tripedal support phases [21,35,36,58].

### 2.3. The Broken Pace (Stepping Pace; Amble)

The broken pace or stepping pace is a four-beat gait wherein the ipsilateral feet are lifted off the ground at around the same time, but the hind foot lands a little before the front foot, yielding an asynchronous or shuffling beat (diagonal/ipsilateral step time ratio of 0.35–0.55). Its lateral advanced placement varies from 0.18–0.13 and diagonal advanced placement from 0.37–0.33 (with increasing speed). The broken pace occurs at a speed of around 2.9 to 6.2 m/s, with a stride duration of 0.61–0.45 s, and a stride length of 2.0 to 3.1 m. It is also a narrow-gauge gait wherein the legs move inward and under the center of the horse’s body and so often has a negative interior straddle value. Like the hard rack or speed rack, the broken pace often leaves trackways wherein diagonal pairs of hoof prints are close together. Though more comfortable than a hard pace due to periods of tripedal and unipedal support, less side-to-side sway, and possessing little vertical motion in the saddle (croup vertical displacement ca. 4.0 cm), the broken pace is less comfortable for the rider than a rack or running walk as there is still some side-to-side motion in the saddle (and the horse’s head may even sway from side to side, though somewhat elevated). The broken pace, like the rack, involves spring mass mechanics and is quite efficient. A horse can maintain a broken pace for some distance, but it can be hard on the horse’s back (which tends to be somewhat ventroflexed or concave) if habitually occurring [19,23]. When compared to the rack of the Icelandic at 3.2 m/s, the mean heartrate of the Mangalarga Marchador horse in a marcha picada gait was lower (106.2 BPM vs. 132.0 BPM), but there was a higher respiratory rate immediately after (ca. 105.0 breaths/minute vs. 70 breaths/minute) and thirty minutes after (ca. 45 breaths/minute vs. 29 breaths/minute) locomoting at 3.2 m/s vs. 5.5 m/s, as well as a higher mean blood lactate level of 3.2 mmol/L (vs. 1.07 mmol/L) [59]. The broken pace is also not as fast a gait as the trot, hard pace, or gallop.

The broken pace is considered undesirable in many of the horse breeds in which it occasionally occurs, such as the Faeroe Island, Icelandic, American and Puerta Rican Paso Fino (i.e., the sobre paso), McCurdy Plantation Horse, and Tennessee Walking Horse.

Yet, the broken pace (stepping pace) is a recognized and desired gait in several horse breeds, whether for transportation or show. In fact, the broken pace is esteemed in Greek horses, such as the Arravani (also known as the Macedonian Pacer); Cretan, or Messara; Peneia; Pindos; Rhodian; and Thessalian, where the gait is called the arravani and performed with elevated front feet; as well as in the Albanian; and the Shan of Myanmar in Southeast Asia, where the gait is called the ahthacha [35,36,60,61,62]. A broken pace is also valued in South African Saalperd, Basuto, and Cape Boer, where the gait is called the kortgang, as well as the South African Nooitgedacht, where the gait is called the styrkstap, and the Dongola of West Africa, where the gait is called the takama [21,35,36,58].

In the Southern United States, the broken pace is a recognized gait in horse shows with the American Walking Pony, Morgan, Saddlebred, National Show Horse, and National Spotted Saddle Horse, where it takes the form of a gait performed at a slower speed than the rack, hence its name, the slow gait. The broken pace is also a favored gait in Spanish-American horse breeds that made their way into Native American hands (whence it is often called the “Indian shuffle”), as in the Appaloosa, Florida Cracker, Nokota, North American Curly, Paint, Sable Island, Spanish Mustang, Spanish Colonial Horse, Tiger Horse, and Walkaloosa [35,36]. On occasion, the broken pace also seems to occur in the Azteca and the Banker and Morab of the United States.

A broken pace is highly esteemed in several South American horse breeds, especially in Peru and Brazil. The Andean and Peruvian Paso are known for their broken pace (the sobreandando), which is ideally performed at a fairly fast speed with overstepping and periods of unipedal support (ca. 8–9%), otherwise it is not very comfortable to the rider in a slower understepping form (aguilillo) [35,36,39]. So too are Brazilian horses, such as the Campeira, Campolina, Mangalarga Marchador, Mangalarga Paulista, Mangolina, Nordestino, Pampa, and Piquira, which display a broken pace known as the marcha picada and which is normally a slower gait (ca. 3.0–4.0 m/s), with a stride duration of 0.93 s and a stride length of ca. 1.8–2.0 m, with extensive tripedal support phases (ca. 60%), but can be performed quicker (stride duration 0.50 s) with periods of unipedal support (ca. 13–19%) [35,36,39,63,64,65,66,67,68,69,70]. A broken pace is also desirable in the Paso Higueyano of the Dominican Republic in the Caribbean, which, in fact, is named after the gait, i.e., the paso higueyano, and is performed in a slow and animated manner with elevated front legs and small steps. A broken pace on occasion occurs in the Marajoara of Brazil and perhaps the Costa Rican Saddle Horse.

Some of the Indonesian horses have been described as having this gait, such as the Sandalwood and Timor of Indonesia, and this seems to be the gait exhibited by the Tongan Singlefooter of Tongan Island, Polynesia.

### 2.4. The Pace (Hard Pace; Flying Pace)

The true or hard pace is a two-beat gait involving heavy coordination of ipsilateral legs which take off and land at nearly the same time (lateral advanced placement of 0.10–0.05, and diagonal advanced placement of 0.45–0.40 as velocity increases). It is performed at a fast speed (3.5–8.1 m/s, up to 10.5–14.0 in the racing or flying pace), with a stride duration of 0.60–0.30 s and a stride length of 2.1 to 3.5 m (up to 5.4–6.3 in the racing or flying pace). The hard pace also involves periods of suspension when all four feet are in the air at once. The hard pace possesses a strong side-to-side motion in the saddle, especially at slower speeds, with the horse’s head held high but often swaying from side to side, and is not very comfortable for the rider [19,23,71,72].

The hard pace is capable of great speeds, even faster than the trot, though not as fast as the gallop, yet it can be hard on the horse as its back is quite concave or ventroflexed and it is metabolically taxing. In theory, gaits with periods of suspension where all four feet are in the air together and then make ground contact, such as the trot, hard pace, and gallop, compress the limb joints more than gaits that maintain ground contact, such as the walk or the alternative lateral gaits of the running walk, rack, broken pace, or broken trot, and in this way can make increased use of the elastic energy stored in the ligaments and tendons of the horse’s legs, and there is less likelihood of ipsilateral feet interfering with each other. A flying pace treadmill simulation in Icelandic Horses at ca. 9.2–12.1 m/s produced very high mean heart rates (205–207 BPM) and mean lactate concentrations (11.9–18.5 mmol/L), with a respiratory rate at 33 breaths/minute after 30 min of recovery [73]. Even higher mean heart rates and/or lactate blood concentrations occurred in the speed pace (ca. 10 m/s) of the Standardbred Horse (heart rate 199–227 BPM; lactate 20.8–20.9 mmol/L) [74,75], less than that of Standardbred Horses in a racing trot at ca. 8.9–11.9 m/s (lactate 15.0–42.7 mmol/L) [76], but approaching or exceeding those of racing Thoroughbreds in the gallop at 15.3–15.9 m/s (heart rate 214–223 BPM; lactate 22.5 mmol/L) [77,78]. Respiratory rates in Standardbreds in a pace at 8.9 m/s (123 breaths/minute) and 10.7–13.2 m/s are less (90–97 breaths/minute) than those in a gallop at 11.3–13.2 m/s (130–135 breaths/minute) [79]. On the other hand, peak ground contract forces are particularly high in the hard pace as compared to the tölt at velocities above 5.0 m/s (10.6 N/kg for front leg and 7.6 N/kg for hind leg in tölt, and 31.1–32.5 N/kg for the front leg in the hard pace and 42.6–44.3 for the hind leg in the hard pace) [40,41,51,72].

The hard pace is particularly well known in the Icelandic Horse, who are able to perform a flying pace or flugskeið at high speeds [19].

The hard pace (along with the broken pace) is also quite common among Turkish breeds of Central and West Asia including the Andolu Yerli, Canik, Lesvos, Mytilene, Turkmene, and West Black Sea Horse, where the broken pace is called the yorga (or the düz yorga when collected) and the hard pace the rahvan, and performed at fast speeds with elevated front legs. The hard pace (along with the broken pace) can also be found on occasion in the Kushum and Yamud (or Iomud) of Turkmenistan and the Kurdi (or Jaf) and Turkoman of Iran [35,36,80,81,82,83,84]. The hard pace or stepping pace likely occurred in the now extinct Turkmene of Turmenistan.

Several breeds of Central Asia can perform the hard pace and broken pace, including the Kalmyk, Kabarda, Karachai, and Kazakh of Kazakhstan and Russia. And the hard pace, along with the broken pace, occurs in horse breeds of the West Asian steppes, including the Karabakh of Azerbaijan, Tushin of Georgia, and on occasion the Altai and Tuva of Russia, as well in the Karabair, Kyrgyz, Yanqi, and Yili of the Eurasian steppes and Tarim Basin in Xinjiang, China. The hard pace also occurs in the Mezen and Vyatka of Russia [35,36,85,86,87,88,89,90,91].

Many East Asian breeds also pace including the Chakouyi, which do so with elevated and flexed front legs, Cheju (or Jeju), Datong, Hokkaido (or Dosanko), Mongolian Wushen (where it is called the joroo), Shan, and Tibetan. The hard pace is also found in the Bhutia (or Yuta), Spiti, and Zaniskari of India [35,36,92,93,94,95,96,97,98,99].

Iberian breeds, such as the Galician and Garrano, sometimes possess a hard pace called the andadura serrada or andadura perfecta and a fast broken pace called the andadura chapeada or andadura imperfecta, and such gaits are found on occasion in the Andalusian and Asturcón. A hard pace is also sometimes found in the French Mérens, German Lewitzer, and Welsh Mountain Pony. And it has made its way into the Andalusian donkey, where it is called the paso de ambladura.

A hard pace also occurs in many harness racing horses that compete in the trot or pace, including the Bulgarian Trotter, Dutch Harness, Finnhorse, French Trotter, German Trotter, Italian Trotter, Spanish Trotter, Orlov Trotter, Standardbred, Nordic Trotter, and Trottingbred.

The hard pace is typically discouraged in South American breeds. In the Peruvian Paso, the fast form of the hard pace with much overstep and suspension is called the huachano (or paso portante or andadura) and is not very comfortable for the rider, though the entrepaso or slower version of the hard pace, performed with less overstep and more lateral bipedal support phases than suspended phases, is a bit more comfortable. In the Mangalarga Marchador or Mangalarga Paulista, the hard pace is called the passo esquipado (or andadura); in the Colombian Paso, the hard pace is called the andadura saltada; and in the Puerto Rican Paso Fino, it is simply called the andadura, where it usually occurs at a fast speed. The hard pace seems to occur on occasion in the Newfoundland Pony but is not too desirable there.

### 2.5. The Broken Trot (Fox Trot)

The broken trot or fox trot is a variant of the trot wherein diagonal limbs lift off the ground in unison (or nearly so), but the front foot comes down just before the hind foot, resulting in diagonal couplets and a four-beat gait. It has a lateral advanced placement of 0.41–0.32 and a diagonal advanced placement of 0.07–0.20 (at increasing velocities). It is still, however, a lateral sequence gait with a footfall pattern of left hind, left front, right hind, and right front. The broken trot occurs at a speed of around 3.2–3.7 m/s, a stride duration of around 0.67–0.58 s, and with a stride length of 2.0–2.5 m. Like the trot, ipsilateral pairs of hooves land near each other in the trackway. In the typical broken trot or fox trot, the forelimbs engage in long sweeping motions with minimal flexion of the carpus, whereas the hind limbs take shorter steps with great hock flexion and springing release of force. Fox trotting horses tend to engage in some head nodding (vertical head displacement of 12 cm), and their back tends to be convex or rounded (dorsiflexed). The fox trot is quite comfortable for the rider (though there is a slight front to back rocking motion along with an occasional small bounce up and down) and is able to be sustained for long distances by the horse [19,23,30,100,101].

Wanderley et al. [59] showed just how energy efficient the broken trot is. They compared the marcha picada (broken pace) and marcha batida (broken trot) of the Mangalarga Marchador at 3.2 m/s, however, and found the marcha batida gait (heart rate 84 BPM; respiration rate ca. 85 breaths/minute; lactate 1.3 mmol/L) was more energy efficient than the marcha picada (heart rate 106 BPM; respiration rate ca. 105 breaths/minute; lactate 3.2 mmol/L) as it had lower average heart rate, respiration rate, and lactate blood levels and compares favorably with trot [59,102,103,104,105]. The broken trot presumably has lower ground reaction forces than the trot or hard pace due to tripedal support phases at slower speeds and maintenance of one foot on the ground at all times, though it cannot reach as high of a speed as the trot, hard pace, or gallop.

The broken trot (or fox trot) occurs in several Siberian horse breeds of the Baikal and Caucasus Mountains, such as the Buryat, Kabarda, Karachai, Transbaikal, and Yakut, where it is called the tropota and involves short quick steps [35,36,106,107,108,109]. The broken trot also occurs in Indian horses that inhabit the Thar Desert, such as the Kathiawari, Marwari, and Sindhi, where it takes the form of a fast gait with periods of unipedal support and elevated front limbs known as the revaal (rewal) or aphcal [35,36,110,111,112]. The broken trot also occurs in some Akhal-Teke horses found in the Karakum desert in Turkmenistan and imported into the United States (where it is called the glide) [113].

In the United States, several southern gaited breeds may possess the fox trot including the Appalachian Singlefoot Horse of North Carolina, which has a fox trot with little front leg elevation and with capping of tracks; Marsh Tacky of South Carolina (the swamp trot or rocking chair trot), which is performed at a slow speed with periods of tripedal or even quadrupedal support for traversing marshes; Missouri Fox Trotter, which has a slower broken trot (flatfoot walk or fox walk) and a faster broken trot (fox trot) with lots of head nodding; as well as in breeds derived from the Missouri Fox Trotter, such as North American Curly and the Walkaloosa [30,35,36,100,101,114,115]. The broken trot can sometimes be found in the Kentucky Mountain Saddle, Morab, National Spotted Saddle, Nokota, Tennessee Walking, and Tiger Horse (glide) of the United States. The park walk of the Morgan is either an animated two-beat slow trot or a slow four-beat broken trot (fox walk), in which case periods of quadrupedal support (c. 5%) are possible [116].

The broken trot also occurs in South American horses. In the Puerto Rican Paso Fino and Puerto Rican Criollo, it can take the form of the paso corto (pleasure or trail gait) at moderate speed and extension, with the head of the horse held high and somewhat elevated forelimbs, and which is very comfortable with periods of tripedal support, or of the paso largo (speed gait) at fast speeds with much extension, with the head held lower along with less front leg elevation, and with phases of hind unipedal support [30]. A broken trot may also occur in the Andean and the Peruvian Paso of Peru, though it is discouraged in the latter breed (where it is called a pasitrote).

Perhaps the broken trot, however, is most famous in Brazilian and Colombian breeds. In Brazil, the Campeiro, Campolina, Mangalarga Marchador, Mangalarga Paulista, Mangolina, Nordestino, Pampa, and Piquira all possess a broken trot (the marcha batida) that can take place with long periods of stance (ca. 65%; stride duration 0.61 s), including tripedal support phases (ca. 27%) or even quadrupedal support phases (ca. 7%), useful in traveling on moist or soft terrain, though it can also be performed at a faster speed (ca. 4.3 m/s; stride duration 0.53 s), with a longer stride length (ca. 2.3 m) with periods of unipedal support (ca. 16%) [30,35,36,63,64,65,66,67,68,69,70]. The marcha batida can be performed in different styles, as a marcha macia with the front hooves kept close to the ground, the marcha batida proper, which is quicker stepping with higher front leg elevation, and the slower marcha boa with tripedal or even quadrupedal support, the favored and most comfortable version. The Colombian Trocha Pura and Trocha y Galope Horse take their names from the broken trot (i.e., the trocha), which most often occurs with quick short steps (having a stride duration of 0.39–0.34 s) [35,36,56]. A similar trocha gait is found in the American Paso Fino Horse through Colombian blood lines. The broken trot occurs on occasion in the Bolivian Paso, Venezuelan Criollo (or Llanero), and Marajoara of Brazil.

Some New Zealand breeds, including the Kaimanawa and Nati, also seem to exhibit the broken trot.

Figure 1 below (see also Appendix A) summarizes the information given above. It presents a modified Hildebrand diagram for symmetrical horse gaits, both natural and alternative lateral ones, contrasting among other things their lateral advanced placement (% of stride duration separating ipsilateral foot contacts), duty factor (% of stride duration hind feet are on the ground), stride duration, tripedal or unipedal support phases, and suspended phases where all four feet are off the ground together.

## 3. Alternative Lateral Gaits in Ancient Art and Literature

The most important evidence for the presence of alternative lateral gaits in early domestic horses occurs in art. Such artistic gait depictions, however, must be carefully interpreted. It is necessary to distinguish intermediate speed alternative lateral gaits from prances (or the raising of a front limb from a standing position), slow walks, and gallops, all of which can look similar at certain stride phases. Indeed, because gait speed is not always apparent in art, an alternative lateral gait is most clearly represented when ipsilateral limbs are high off the ground at the same time. For this reason, several famous images of laterally gaited horses are quite ambiguous.

Certain depictions of horses in the Lascaux caves of France (ca. 17,000 BC) have been interpreted as engaged in the alternative lateral gait of a hard pace or rack. However, it is hard to tell whether limbs are or are not on the ground, and such depictions are more likely to represent galloping horses as with other horse images in the Lascaux caves [117] (pp. 7, 26) and [118,119,120]. Two other early horse depictions—a horse figured on the tomb of Rekhmire in Sheikh Abd el-Qurna, Thebes (TT1100), ca. 1471–1448 BC (their Figure 3 in [121]), and a bas-relief in the North Palace of Nineveh in Assyria (ca. 645–635 BC) of a horse fleeing a predator (their Figure 3 in [122])—could be interpreted as displaying a running walk gait, since the hind limbs seem to extend far underneath the horses’ bodies while the ipsilateral front feet are in the air. It is hard to be certain, however, that the artists intended to depict more than a standard walk. Most likely they did not.

A coin from ca. 492–480 BC depicts Alexander the Great on a Macedonian horse with what might be two ipsilateral feet off the ground at the same time (their Figure 46 in [123]), as does a coin from ca. 440 BC of the Thracian king Sparadokus (their Figure 16 in [60]). These images may well portray horses in a racking gait, since the hind hooves seem to be close to landing or just making contact while the ipsilateral front hooves remain quite elevated. However, it is hard to be sure that an intermediate speed racking gait is intended as opposed to a slow walk with animated front legs or a prance. Other images of Alexander, in fact, feature him on a trotting horse, and depictions of rulers on prancing horses with an elevated front leg become commonplace in ancient art (as noted in Xenophon’s Περὶ ἱππικῆς 11 of ca. 355 BC) [124]. The famous equestrian statues of Marcus Nonius Balbus in Herculaneum (ca. 25 BC), the Horses of St. Mark (Cavalli di San Marco) in St. Mark’s Basilica in Venice, Italy (ca. 200 AD), and the bas-relief (MNHA 261) of the Celtic goddess Epona from Dalheim, Luxembourg (ca. 200 AD) also likely represent nothing more than a walking gait with elevated and animated front legs as in the Spanish walk of the Andalusian or park walk of the Morgan Horse or a prance wherein a horse is raising a front leg from a standing position [123,124,125].

Finally, the West Frieze of the Parthenon, sculpted around 440 BC, depicts a procession of riders on horseback transporting the peplos (robe) of the goddess Athena from Kerameikos to the Temple of Athena Nike on the Acropolis during the Panathenaic festival. Block W-9 in particular contains a carving of a horse that is often said to display an alternative lateral gait with two ipsilateral limbs off the ground together in a collected posture (their Figure 49 in [123], their Figure 3.19 in [124], and [126]). Yet the image more likely represents a horse being reined to a stop from the gallop, as with the horse behind it, since its weight appears to be shifted posteriorly.

Turning to literature, early references to alternative lateral horse gaits are also quite ambiguous. The distinction between driving (penna) and chasing (parh; lahlahhiskinu) gaits in the Hittite Kikkuli horse training manual (ca. 1450 BC) is no longer thought to distinguish between trotting and pacing gaits but rather between trotting and galloping gaits or different speeds of a walking gait [127,128].

There are suggestive references, however, to what may be gaited horses in Roman authors, albeit to breeds located in Greece, Spain, and Persia. Lucilius, in his *Saturae* 476 (ca. 130 BC), describes the horses of Lusitania in Spain as possessing optimal gaits (gradarius optimus vector). Virgil, in his *Georgics* 3.117 (29 BC), talks about horses from Thessaly in Greece employing “distinguished balled steps” (gressus glomerare superbos). Strabo, in his *Geographica* 3.4.15 of ca. 5 BC, notes that horses brought to Iberia by the Celts, i.e., Celtiberian horses (Κελτιβήρων), move swiftly and easily (ταχεῖς καὶ εὐδρόμους), just like those of Persia, and are accustomed to climbing mountains and bending their limbs at bidding when required (κατοκλάζεσθαι ῥᾳδίως ἀπὸ προστάγματος, ὅτε τούτου δέοι). In fact, we know that such horses were employed by Celtiberian mercenaries hired by Athens and Sparta in the battle against Thebes of 369 BC (described by Xenophon in his Ἑλληνικά 7.1; see also Diodorus Siculus, *Bibliotheca historica*, 5.29–33).

Petronius, in his *Satyricon* 86 (ca. 60 AD), also speaks of the great Asturian-like horses in Macedonia (asturconem Macedonicum optimum). Seneca, in his *Epistolae morales* 87.10 (ca 63 AD) contrasts simple horses (caballus; equus) with Spanish Asturian horses (asturconibus) and trotting horses (tolutariis). Pliny the Elder, in his *Naturalis historia* 8.57 (77 AD), writes of horses from Galacia and Asturia in Northern Spain (larger theldones and smaller asturcones) that had an unusual gait (non vulgaris) that was supple (mollis) and involved the successive uncoiling of balled limbs (alterno crurum explicatu glomeration), based on which (other) horses are taught to adopt a speedy trot (tolutim carpere incursum). Similarly, Silius Italicus, in his *Punica* 3.336 (84 AD), notes that the Spanish Asturian horse travels in “balled steps leaving the (rider’s) back unshaken” (inconcusso glomerat vestigia dorso). Martial, in his *Epigrams* 14.199 (85 AD), goes on to describe how the Asturian horse “picks up its hoof in a quick rhythmic manner” (ad numeros rapidum qui colligit unguem). And later, Vegetius, in his *Digesta artis mulomedicinae* 1.56 and 3.6 (ca. 430–435 AD) writes of the Persian saddle horses which possessed splendid gaits of great value (incessus nobilitate pretiosos), namely, intermediate gaits (ambulatura media) between those of trotters (tolutarios) and gallopers (totonarios). In particular Persian horses have gaits (ambulaturae) with short and quick steps (gradus … minutus, celer), which delight and excite the rider (qui sedentem delectet et erigat). For they are taught to walk at a trot (tolutim ambulare) supplely (molliter) and in a light and flattering manner (levitatem et quaedam blandimenta vecturae), with short steps (minutos gressus; minutum ambulans), elevated legs (altius crura), and bent knees and hocks (inflexione geniculorumn atque gambarum), i.e., in a manner similar to that of Spanish Asturian horses (or asturconibus).

However, it has been disputed whether these descriptions relay no more than that these Greek, Spanish, and Persian horses moved with elevated front legs in a high-stepping and collected trot, as with the paso de Andadura of the Spanish Andalusian; the paso nadado (paso español) of the Portuguese Alter Real, Giara, and Chilean horses; or the park trot of Hackney and Morgan Horses [129,130,131,132,133,134]. The Latin noun glomeratio and verb glomerare are ambiguous in meaning (translated above as balled), referring either to something which is rolled-up or rounded (i.e., flexed joints) or more abstractly to something that is assembled or joined together (i.e., collected or coordinated steps). The only text above that seems to definitively pick out an alternative lateral gait, as opposed to a collected and elevated trot, is that of Silius Italicus, who noted that the gait of Asturian horses leaves the rider’s back unshaken, perhaps explaining the frequent use of the word mollis (fine or smooth) in describing such gaits in other authors. Indeed, Xenophon’s *Περὶ ἱππικῆς* 7 (ca. 355 BC) only mentions the gaits of walk, trot, and gallop in Athenian horses, but this may be because he omitted the gaited horses located in more remote parts of Greece.

Still, the contrast between trotters (tolutarios) and horses with other gaits (ambulaturae; glomerare) in these classical Latin texts does seem to indicate the presence of some sort of ambling gait in the Spanish, Persian, and Greek horses of the Roman era (ca. 130 BC–435 AD). And it is hard to explain otherwise how the gaited sixteenth-century Iberian horses, from which the gaited breeds of the Americas derived, gained such alternative lateral gaits. In fact, certain combined literary, pictorial, and historical evidence strongly suggests that early Celtiberian, Greek, and Persian horses possessed alternative lateral gaits, and that they derived from importation of Central Asian horse breeds.

Justin’s *Historia Philippicae* 9.2 (ca. 175 AD), itself based upon the lost *Liber Historiarum Philippicarum* (ca. 10 AD) of Pompaeus Trogus, mentions fine mares (nobilium equarum) that Phillip of Macedonia brought back from the Ferghana Valley of Uzbekistan, then part of the Greco-Bactrian Kingdom (formerly Scythian and Persian) in 339 BC. Chinese historical texts a few centuries later will mention similar “heavenly horses” (tianma) imported into China in 104 BC from the Ferghana Valley during a war with the Greco-Bactrian Kingdom of Uzbekistan (*Shiji* 24; *Hanshu* 96) [135,136,137].

In addition, we find clear depictions of pacing horses in reliefs and ceramic statues of China during the Han dynasty (ca. 25–220 AD) [138,139,140,141,142], namely, the chariots depicted on stone slabs from the Wu Family Shrine (ca. 147 AD) at the Wuzhaishan Site (North Slope, East Wall) in Shandong Province (ca. 147 AD) or the Chulan Tomb 2 (ca. 171 AD) from Suxian, Anhui Province, and the well-known Flying Horse of Gansu statue (ca. 220 AD) from the tomb of General Zhang in the city of Wuwei, Gansu Province, China (their Figure 45 in [117], their Figure 1 in [143], and their Figure 3a–c in [144]). Elsewhere in early Oriental art and literature, a Turkestan painting from ca. 700 AD shows a horse and camel pacing side by side, and the Turkish Şine-Usu Inscription in Mongolia (ca. 747 AD) seems to mention a pacing (yorga) horse race on the seventh column of its south side [16] (p. 293) and [145,146].

For whatever reason, in the West distinct evidence for alternate lateral gaits in horses only shows up in the Middle Ages and Renaissance. Pictish (Celtic) carvings on stone slabs unmistakably show Scottish soldiers on racking horses (ca. 800–900 AD), such as those from Meigle, Pershire (now in the Meigle Museum), the Cross Slab in Edderton, Easter Ross, and the Hilton of Cadboll Stone (now in the National Museum of Scotland, X.IB 189) (their Figure 3 in [147]). The royal seals of King Richard I (1197) and John (1215) of England, Normandy, and Aquitaine, portray them mounted on racking horses on the reverse side (now in the British Museum 2000.0103.6; 1987.0103.1). Also perhaps depicting a racking horse (though less distinctly) is an icon of St. George on a horse with the youth of Mytilene from the Near East (ca. 1250).

These artworks are matched by explicit descriptions of alternative lateral horse gaits in medieval literature. The cleric William Fitzstephen, in his *Descriptio nobilissimae civitatis Londoniae* 11 (ca. 1172), describes a horse market located in a field (West Smithfield) outside of London. Fitzstephen found it a joy to behold there gaited horses (gradarios) that sweetly ambled (suaviter ambulantes) by alternatively raising and lowering the legs on the same side of the body in unison (pedibus lateraliter simul erectis quasi et subalternis et demissis), in addition to rougher (trotting) horses (durius equos) that raised and lowered their opposite front and hind legs together (a contradictoriis pedes simul elevant et deponent), and swift (galloping) horses that first throw out both front feet followed by both back feet (pedibus anteriorobis simul solo … et posterioribus similiter). The German Dominican Albert the Great, in his *De animalibus* 22.54 (ca. 1260), describes four gaits of horses: the walk (peditatio), trot (trotatio), gallop (cursus), and amble (ambulatio). He notes that in the amble (ambulatio), the horse moves by simultaneously lifting up the front and hind foot on the same side of the body (simul in eodem latere unum anteriorem et unum posteriorem leuat pedem). And he claims it occurs more sweetly (suavius) if the horse does not elevate its legs too much, and places the front feet on the ground more quickly than the hind foot (aliquantulum citius anteriorem quam posteriorem figit pedem)—perhaps describing a rack with elevated and animated front legs.

Such alternatively gaited horses (i.e., Palfreys, Galloways, Hobbies) seem to have become quite popular in the later Middle Ages as several illuminated manuscripts depict clerics, knights, and nobles riding in a racking gait including an Apocalypse manuscript of ca. 1275 (now in the British Library, Add MS 35166), the English *Queen Mary Psalter* (ca. 1310–1320) and Scottish *Taymouth Hours* (ca. 1325–1335), the French *Très riches heures du Duc de Berry* (ca. 1410–1416), along with Devonshire Hunting Tapestries of ca. 1425–1450 (now in the British Library, Royal MS 2.B.VII.f151b) [117] (pp. 173, 177) and [148] (pp. 7, 149). The Ellesmere *Canterbury Tales* manuscript (ca. 1400–1410) and the Italian Fiore dei Liberi’s *Flos duellatorum* (ca. 1410) depict horses in what may be a running walk with hind legs extending far underneath the horses’ bodies. The city seal of Pavia, Italy (ca. 1450), in fact, contains a clear illustration of a laterally gaited horse (their Figure 45 in [124]).

References to alternatively gaited horses are also quite common in the East during the Fourteenth to Sixteenth Centuries in the Mamluk Empire [146,147,148,149,150]. Abou Bakr Ibn Badr, in his *Nâçerî* 19 (ca. 1333) written for a Mamluk sultan, details ten forms of ambling, in contradistinction to the trot, found in Arabian horses, mules, and camels. Important are his contrasts between three velocities of the two-beat hard pace, the slow amble (mekhâm), pleasure amble (harwalah or hemledjeh), and speed amble (rahwanah), along with the four-beat amble (rakd), presumably the rack or perhaps the broken pace. Another Mamluk manual of horsemanship, the *Nihāyat al-su’l wa-al-umnīyah fī ta‘allum a‘māl al-furūsīyah* (1371), depicts Mamluk warriors using spears or lances while riding on gaited horses [151,152]. Elsewhere, an illustration from the Mughai School in Oudh, India, of ca. 1675 (now in the Victoria and Albert Museum, IS.133:31/B-1964) depicts a prince practicing falconry on an ambling horse [117] (p. 180) and Asvasastra’s *Treatise on the Nature and Illnesses of Horses* (ca. 1750) shows horses performing a broken pace (or rack) at the Newar royal court in the Kathmandu Valley of Nepal.

The distinction between trotting and ambling horses became commonplace in the Western Renaissance, with the terms gradarius and ambulatura (ambladura) associated with intermediate speed alternative lateral gaits, even if they had broader usages in the ancient world and could mean simply a walk, trot, or gait, in addition to perhaps an amble [133]. Polydore Virgil, in his *Anglica Historia* 1.15 (1534), notes that many English horses did not trot but instead rather paced (non succussat sed graditur). Similarly, Thomas Blundeville’s *The Arte of Ryding and Breakinge Great Horses* (1560) contrasts horses that “have a trotting pace, as the mares of Flanders and some of our own mares” with “ambling horses, to travel by the way … (such as) a fair jennet of Spain, or at least a bastard jennet, or else a fair Irish ambling Hobbie”, and with “swift runners … (such as) a horse of Barbary or a Turk”.

Gervase Markham, in his treatise *Cavelarice, or The English Horseman* 4.1 (1607), delineates three intermediate speed horse gaits, the certain amble (thorow amble; certaine amble), the uncertain amble (traine; racke; incertaine amble; shuffling and broken amble), and the trot. The amble in general is denoted as a two-beat hard pace, or “the taking vp of both the legs together vppon one side (he must take vp his right fore foote, and his left hinder foote), & so carrying them smoothly along, to set them downe vpon the ground euen together, and in that motion be must lift and winde vp his fore foote some what hye from the ground, but his vnder foote he must no more bvt take from the ground, and as it were sweep it close by the earth”. By way of contrast, “when a horse trots, he takes vp his feet … to which is crosse wise, as the left hinder foote & the right fore foote”. Markam goes on to favorably describe the certain amble, whether by nature or training, as one in which the horse passes over a sizeable quantity of ground in a few paces, with smooth, certain, and deliberate steps. On the other hand, the uncertain amble is undesirable and occurs in disordered or weary horses when the horse performs a pace with short, quick, and busy strides, taking up the feet on the same side “thicke and rouddly together”, and traveling only a short distance in a long time. Later equestrian works, such as William Browne’s *The Arte of Riding the Great Horse* (1628) and William Cavendish’s *A General System of Horsemanship* (1658), offer similar contrasts of the trot, hard pace, and broken pace.

## 4. Evolutionary Origin of Alternative Lateral Gaits in Fossil Equids

Alternative lateral gaits are rare in mammals, typically occurring only in longer-legged species such as camels, alpacas, llamas, guanacos, vicuñas, giraffes, okapis, gerenuks, elephants, bears, bandicoots, maras, the maned wolf, some longer-legged dog breeds (including greyhounds, bloodhounds, Great Danes, Rhodesian ridgebacks, salukis, and weimaraners), and of course the modern domestic horse [153].

Horses (equids), along with tapirs and rhinoceroses, belong to the Order Perissodactyla. Horses (family Equidae) emerged in the early Eocene around 56 Ma [154,155,156]. The first equids (hyracotheres) were small (around 35 cm in height at the withers) and tetradactyl with four toes on the front foot (manus) and three toes on the hind foot (pes), locomoting in a subunguligrade posture with body weight distributed over several small hooved toes and a large padded foot, much like today’s tapirs [154,155,156]. Modern non-equine perissodactyls, i.e., tapirs and rhinoceroses, have a similar foot structure and seem to exclusively employ the trot as their intermediate speed gait, and this presumably represented the locomotive behavior of basal perissodactyls including equids. Tridactyl anchitheriine equids evolved longer legs with a prominent central toe and reduced lateral toes. At some point, perhaps in the late Oligocene or early Miocene, anchitheriines such as *Anchitherium* seem to have lost their footpad entirely and locomoted in an unguligrade posture with their weight centered over a large hoofed central toe assisted by smaller lateral toes. Stem-equine anchitheriines of the early Miocene such as *Parahippus* developed long proximal central phalanges with enlarged plantar V-scars and so most likely a spring foot that could take advantage of the elastic rebound energy provided by suspensory ligaments after ground impact [157,158]. Such changes coincide in part with the spread of grassy plains (ca. 24 Ma) and seem to represent adaptations to life on a flat and somewhat pliable substrate (at least in comparison to rocky or hilly terrain), as well as to more open terrain with an increased need for efficient long-distance migration as well as quick evasion of ambush predators, if not pursuit predators which only evolved later it appears [159,160,161].

At some point in their evolutionary history, perhaps beginning with tridactyl members of the subfamily Equinae, equids gained the ability to avail themselves of intermediate speed alternative lateral gaits. For there is evidence from tridactyl equid trackways that perhaps some Miocene and certainly some Pliocene equids employed the alternative lateral gait of a rack (and perhaps a running walk), in addition to the diagonal gait of the trot and the asymmetrical gait of the gallop seen in hipparionin equid trackways from the late Miocene and Pliocene of Italy (ca. 6.0–5.3 Ma) and Spain (ca. 8.7–4.2 Ma) their Appendix A in [162]. For a trackway of *Scaphohippus* from the lacustrine Greer Quarry site (ca. 14.5 Ma) near Barstow, California, United States, while incomplete, is very similar to those made by modern racking breeds, even if a trotting gait cannot entirely be ruled out [163], and trackways laid down by *Eurygnathohippus* in ash at Laetoli Site G in Tanzania, Africa (ca. 3.7 Ma) most definitely display racking gaits, along with a possible running walk gait [162,163,164]. There is no way to know at the moment, however, if such fossil equid made use of the same neural and genetic pathways for these alternative lateral gaits as modern horses do.

Miocene and Pliocene equids, excepting some hipparionins, were shorter than today’s horses (ca. 80–95 cm in height at the withers in the early to middle Miocene and ca. 100–130 cm in height in the late Miocene through Pliocene), but they were still long-legged mammals adapted for rapid travel over diverse substrates through the use of various gaits. Alternative lateral gaits would have been beneficial for such equids for a couple of reasons. In the first place, the joints and ligamentary systems of tridactyl horses seem to have been less restrictive than those of modern horses and so the limb joints were less stable and more prone to hyperextension [157,158,159,163,164]. Not only would the lateral hooves have helped with stabilizing the joints and preventing hyperextension by reducing the force of hoof impact, but so too would alternative lateral gaits, such as the rack, running walk, or broken pace, with their periods of tripedal support as well as continuous unipedal support. Secondly, as alternative lateral gaits allow for maintaining ground contact at all times, even at intermediate speeds, as well as increased periods of tripedal support, they provide for sure-footedness on slippery or highly deformable substrates, such as ash, sand, mud, or wet rocks (where such equid trackways were, in fact, preserved), as well as on uneven and sloping terrain, sometimes littered with rocks, branches, and stumps. Thirdly, as we have seen some of the alternative lateral gaits such as the running walk and broken trot may equal or exceed the trot in metabolic efficiency. A final intriguing but less likely possibility is that alternative lateral gaits (especially the hard pace which is quicker than the trot) could have helped Miocene equids avoid attacks of ambush predators, along with the even faster gallop.

Interestingly, at some point (likely with the Monodactyl equini), the ability of equids to perform alternative lateral gaits was lost. Alternative lateral gaits do not occur in most horse breeds today (*Equus ferus caballus*), including some of the more ancient breeds such as the Exmoor, Sorraia, and most Arabian horses, along with the likely non-domesticated Przewalski’s horse (*Equus przewalskii*), which split off from modern horses ca. 45,000 years ago [165]. Nor are alternative lateral gaits known in wild asses (subgenus *Asinus*), namely *Equus africanus*, *Equus kiang*, and *Equus hemionus*, or in wild zebras (subgenus *Hippotigris*), namely *Equus grevyi*, *Equus quagga*, and *Equus zebra*, groups which split off from modern horses (*Equus ferus caballus*) ca. 4.5–4.0 Ma [166,167].

Different explanations have been offered for the equid loss of alternative lateral gaits and the transition to trotting (and galloping) gaits. The classic explanation was that as equids moved onto open and flat arid grasslands, high speed gaits were needed to flee from predators, favoring longer legs and the monodactyl foot. In contrast, tridactyl horses were adapted to locomotion on the unlevel and uneven terrain of woodlands and forests where the extra digits and alternative lateral gaits (with increased periods of tripedal support and continual unipedal support with the exception of the hard pace) aided in traction, joint stability, and agility [157,158,159]. There may be some truth to this, and short bursts of speed were likely useful in avoiding predation on the grasslands, but, as has been pointed out, horses seem to have evolved long limbs and reduced digits well before high-speed pursuit predators came on board and monodactyl and tridactyl forms coexisted in many biomes [160,161]. Hence a more recent view is that the development of monodactyly, and the transition from alternative lateral gaits to the diagonal trotting gait, was an adaptation for migration between patchy resources in open, level, grassland environments occurring around 10 Ma. For, as we have seen, biomechanically, a trotting gait, making use of the elastic energy provided by the suspensory apparatus of a spring foot, is more efficient than a racking gait for long-distance travel at intermediate speeds. Moreover, Miocene to Pliocene equids of the stem-equine Anchitheriinae and Equinae not only possessed a dominant middle (third) digit encased in a keratinous hoof (or a single toe) but also seem to have developed a spring foot with a ligamentary suspensory apparatus allowing for release of elastic energy derived from ground contact and the ability to sleep while standing and being ready to feel predators on short notice (again likely related to an open habitat) [157,158,159]. And the gallop, allowing for the highest speeds in short distances, would be most useful in escaping predators (whether ambush or later pursuit ones) on open terrain.

Alternative lateral gaits then seem to have been present in some tridactyl equids from ca. 24–3 Ma but were lost in the monodactyl equids that arose ca. 10 Ma.

## 5. Origin and Spread of Alternative Lateral Gaits in Modern Domestic Horses

The monodactyl genus *Equus* derived from other monodactyl equids such as *Dinohippus* ca. 5.5–4.5 Ma. By 400,000 years ago, the last tridactyl horse species were extinct, and by 11,000 years ago, *all* North American horses were extinct. Three, somewhat isolated populations of *Equus ferus caballus* coalesced on the Continent: one occupying the Central Asian steppes, another existing in Iberia and separated from Europe by the Pyrenees, and finally an East Asian population in Mongolia and China separated from the rest of Asia by the Altai Mountains and the Taklamakan and Gobi deserts [168]. It is from these horses that domestication occurred.

The latest evidence suggests that the domestication event from which modern horses sprang did not occur, as previously thought, ca. 3500 BC within the Botai region of Kazakhstan in Central Asia and then spread via the Yamnaya culture into Central and Southern Europe ca. 3000–2500 BC, but occurred rather ca. 2200 BC within the Volga-Don region of Western Russia and then spread via the Sintashta Culture of the Ural Mountains and Pontic-Caspian steppes of Eastern Europe and Central Asia [169,170,171,172]; for a contrary view, see [173]. Moreover, modern genetical studies have tied the development of (most) alternative lateral gaits to a particular allele (A) of the DMRT3 gene that arose between 9600 and 1200 years ago [174,175]. So, alternative lateral gaits, whether based on novel genetic, neural, and physiological mechanisms, or representing reversion to an earlier state, likely reappeared in horses ca. 3500–2200 years ago just before or after the horse’s domestication. From there, various cultural groups seem particularly important in the spread of gaited horses, especially the Sintashta, Scythian, Celtic, Greek, Persian, Turk, Mongol, and Spanish.

It is quite possible that the very first culture to domesticate horses, the Sintashta Culture of Russia and Kazakhstan, known for their horse-drawn chariots, possessed alternatively gaited horses and helped to spread them through Central Asia ca. 2050–1750 BC [170]. For local Kushum and Kazakh horses of Kazakhstan possess the DMRT3 gene and, at least with the latter breed, alternative lateral gaits [88,89,90,92]. But it seems to have been the Scythians, also known for their horse-drawn chariots, that really spread alternatively gaited horses throughout Central, Southern, Northern, and Western Asia, as well as Eastern Europe. The Scythians expanded from the Eurasian Steppe and Altai Mountains of Southern Siberia (ca. 900–800 BC) into the Pontic-Caspian Steppe of Central Asia (ca. 800–700 BC), into the Tarim Basin of Northern China ca. 800–700 BC, and into Western Asia (ca. 600 BC), before being conquered by the Persians in ca. 300 BC and by the Macedonians in ca. 339 BC [176,177,178,179]. Regions of Asia previously inhabited by the Scythians have gaited horses to this day including the Altai of Mongolia (and Russia), the Kabarda and Tushin of Georgia, the Kalmyk, Karachai, Tuva, and West Black Sea Horse of Russia, the aforementioned Kazakh of Kazakhstan, the Kyrgyz of Kyrgyzstan, and the Yanqi and Yili of Xinjiang, China. More specifically, the hard pace (along with the broken pace) occurs in horse breeds of the West Asian steppes occupied by the Scythian Cimmerians (800–400 BC) including the Karabakh of Azerbaijan, Tushin of Georgia, the Altai and Tuva of Russia, as well as in regions associated with the Scythian Saka (700–200 BC) of the Eurasian steppes and Tarim Basin in Xinjiang, China, including the Karabair, Kyrgyz, Yanqi, and Yili breeds. Scythian horses, in fact, were the very ones famed in the Ferghana Valley of Uzbekistan (ca. 400 BC) and imported into Macedonia by Philip in 339 BC and into China by the Han Dynasty in 104 BC, and they may have been imported into the Mediterranean even earlier by the Archaemenid Persians. It is thus likely that the earliest gaited horses of Greece (Pindos, Rhodian, Thessalian) and China (Chakouyi, Datong, Gansu, Wushen), known from art and literature, were all derived from Scythian ones. The establishment of the Silk Road ca. 200 BC during the Han Dynasty between Egypt, Greece, India, Asia, and Europe, which presumably allowed exchange of livestock including horses, further helped to spread Scythian gaited horses to areas such as India (perhaps having gaited horses even earlier through immigration of Indo-Iranians) and Southeast Asia—including into Mongolia under the Xiongnu Empire (209 BC-100 AD), into Korea during the Kofun Period (300–538 AD), into Tibet through the South Silk or Ancient Tea Horse Road during the Tang Dynasty (ca. 618–907), and later throughout all of China via a horse market set up in Tianzhu County during the Ming Dynasty (1368–1644 AD), and finally into Hokkaido, Japan ca. 1600 [97,180,181]. There also seems to be a link between the ancient Java horse and the broken pacing gaits of the native South African breeds [35,36].

There is also a close connection of gaited horse breeds with the cultures of the Sassanid Persians (ca. 700–900), Oghuz Turks (ca. 766–1055), and Seljuk Turks (ca. 1037–1308), as seen in depictions of gaited horses in Turkish art and literature of the eighth-century, and the presence of gaited horse breeds (usually with a broken pace or hard pace) in regions formerly inhabited by the Persians and Turks including the extinct Turkmene and Yamud of Turkmenistan, the Karabair of Tajikistan and Uzbekistan, the Karabakh of Azerbaijan, the Andolu Yerli and Canik of Turkey, and the Turkoman of Iran [80,97,172,182]. Finally, the Mongols (ca. 1206–1294) also seem to have helped spread gaited horses throughout Asia. In fact, Wallner et al. [183] point out that European horse breeds cluster together in a roughly 700-year-old haplogroup (ca. 1300) likely transmitted to Europe through the importation of two lineages of Oriental stallions, namely, Arabian and Turkoman. This history of the spread of gaited horses helps explain why there are strong genetic links between the Mongolian and Kazak horse breeds and the Tuva, Turkmene, Tibetan, Kyrgyz, Bhutia, Zanskari, Chakouyi, and Hokkaido [97,181]. There is also a close genetic connection between Caspian and Anatolian horses and gaited Greek horses (Pindos; Cretan) as well as gaited Turkish horse breeds (Kurd, Akhal Teke, Turkoman) [60,182,184]. The Pindos breed in particular shows genetical similarities to Central Asian gaited horse breeds (i.e., the Kyrgyz, Tushin, and Altai) [184]. So there was likely an Anatolian corridor for equine gene flow between Asia and Europe [185].

When it comes to the spread of gaited horse breeds in the West there is a close connection with the Celts. The Celtic Hallstatt Culture, known for its horses, arose in the Alps of Austria and Switzerland (ca. 800 BC) and Celtic peoples seem to have migrated into Iberia (ca. 700–600 BC), Ireland (ca. 500–400 BC), Wales (ca. 450 BC), Yorkshire (ca. 450 BC), and Scotland (ca. 400 BC). It is most probable that the Celts, who had spread east to the Black Sea by ca. 450–350 BC, made contact with Scythians and imported gaited horses directly from them (coinciding with the war chariots that show up in Celtic graves). Another possibility is that the Celts imported gaited horses from Persia or Greece after they had invaded Greece and Macedonia and settled in Galatia (ca. 298–277 BCE), but such horses would have needed to make their way throughout Spain, France, and Britain fairly quickly before Celtic power was compromised by the Romans in 57–19 BC and the Anglo-Saxons in 440–775 AD. In any case, many areas of Celtic inhabitation have had or currently have gaited horse breeds, namely, the extinct Galloway Pony of Yorkshire and Scotland, the Welsh Mountain Pony of Wales, the extinct Hobby Horse of Ireland, the extinct Bidet Briton of Brittany, France, the Mérens of the French Pyrenees, the French and English Palfrey, the extinct Celtic Asturcón, and the Castilian (i.e., Spanish Jennet), Galician, and Garrano of Spain and Portugal. And, as we have seen, historical writings of the Romans suggest that Celtiberian horses were gaited by at least 130–129 BC (if not earlier), Pictish carvings from the Celtic Strathclyde Kingdom (ca. 800–900 AD) depict gaited horses, and DNA from Celtic (or perhaps Viking) horses located in the Northern Yorkshire region of England (ca. 850–900 AD) was found to possess the DMRT3 A-allele common to gaited breeds [174]. Unfortunately, only a couple of Southern European horses from 200 BC–800 AD have been tested so far, and none of them possessed the DMRT3 A-allele [174,175]. Given all of the evidence, then, it seems there was a lineage of Celtic gaited horses in France and Spain going back to 700–500 BC, which then spread into Britain (ca. 500–400 BC) and which was derived from Scythian or Greek gaited horses. Yet so far little genetic connections have been found between Celtiberian and Asian gaited horse breeds, perhaps due to dilution of genetics by mixing with local breeds. Moreover, such Celtic horses seem to have favored the alternative lateral gaits of the rack and running walk as opposed to the hard pace of Asian breeds, as evidenced by the spread of such gaits into the New World via Iberian horses, and the depictions of the racking gait in English Palfreys, French Haquenées, and Pictish Galloways in medieval art [147,148,149].

One particular puzzle, in fact, is how the rack and hard pace got into Icelandic horses, of whom ancient Icelandic horses living from 850–1050 AD were found to possess the DMRT3 A-allele [174]. Icelandic horses share a gene pool both with certain English and Scandinavian horses (Shetland, Norwegian Fjord, and the gaited Scottish Highland and Welsh Pony) as well as with Mongolian horses, though no clear link with gaited Spanish breeds [186,187,188]. Hence it is most likely that the Vikings introduced gaited horse breeds of Asian origin into Iceland and the Faroe Islands (as well as perhaps Ireland and Scotland) via Scandinavia ca. 850 AD [174].

Another possibility, however, is that the Vikings imported gaited Celtic horses from Britain into Iceland. Still, there is direct and indirect evidence that gaited Celtic horses made their way into Britain quite early; they are associated with Pictish monuments (ca. 800–900 AD), and horses of York from ca. 750–850 were found to possess the DMRT3 mutant A-allele [174], which in turn may have led to the medieval Scottish Galloway and Irish Hobby breeds. A horse from Beauvais in the Hauts-de-France region of northern France (ca. 1450–1500) was also found to possess the A-allele of the DMRT3 gene suggesting it was gaited [172]. Moreover, as Wutke et al. [174] note the DMRT3 A-allele is strikingly absent from both ancient and modern Scandinavian horse breeds. Currently, the earliest presence of the DMRT3 A-allele associated with Asian gaited horse breeds occurs in a medieval horse from Tavan Tolgoi in the Ömnögovi Province of Mongolia (ca. 1200–1370) [172]. However, this will probably be pushed back much earlier as more horse samples are tested from earlier periods in the East and West, especially from 200 BC–800 AD [174,175]. So, there is still a puzzle whether gaited horse breeds made their way up into Iceland from Celtic lineages in Britain, or down into Britain from Asian lines in Iceland, or perhaps some combination of the two wherein Celtic and Asia gaited horse lineages were mixed together in Europe. Little evidence, however, supports the view of Wutke et al. [174] that the Vikings spread the gaited allele east via incursions into Western Asia.

Lastly, the Portuguese and Spanish imported many gaited horses (Spanish Jennets derived from Celtic breeds) into the New World during the sixteenth century. The rack is common to many horse breeds of Spanish descent in the Southern United States. Modern genetics tends to confirm the tradition that such gaited horses originated from Spanish blood lines that made their way into the Southern United States and Mexico, and also into New England with the Narragansett Pacer, and were later crossed with other breeds such as the English Thoroughbred [186,188,189,190]. For there are close genetic links between North American gaited breeds (such as the Saddlebred, Florida Cracker, Banker, and Marsh Tackey) and Spanish breeds such as the Andalusian, Pura Raza Galega (or Galician), Barb, and Lusitano [186,191,192,193]. A rack is also common to the Paso Fino breeds of Columbia and Puerto Rico that also had a Spanish derivation.

There is also a close connection of Celtic and Spanish horses with the running walk gait. As we have seen, some of the medieval paintings of Palfrey horses from England, France, and Italy, seem to depict a running walk, in particular those showing knights sparing on horseback: perhaps the lack of vertical motion in the saddle felt in the running walk gait may have made balancing and striking easier on horseback. And the Tennessee Walking Horse, Montana Travler, Canadian Pony of the Americas, Sable Island Horse, and Walkaloosa of North America, the Peruvian Paso of South American, and the Abaco Barb of the Spanish Caribbean, all of which perform some variant of the running walk, have their origin with Iberian horse lineages. Perhaps some combination of the Galician, North African Barb (Moroccan), and Castilian (Spanish Jennet) horse of southern Spain, for the name of the running walk in South America (i.e., paso llano) seems to derive from an original label of “paso castellano” and so association with horses of Castile. It is said to be a gait well-adapted to a coastal or desert environment.

Somewhat mysterious in origin are the five or so groups of horses that display a broken trot, namely, those in Siberia, Russia (Buryat, Transbaikal, Yakut), India (Kathiawari, Marwari, Sindhi), Turkmenistan (Akhal-Teke), North America (Appalachian Singlefooter, Carolina Marsh Tacky, Missouri Fox Trotter, Nokota, and North American Curly, Tiger Horse), New Zealand (Kaimanawa, Nati), and those in South America (the Campeiro, Mangalarga Marchador, and Nordestino of Brazil, and the Columbian Trocha). There may be an early connection with Andronovo and Indo-Iranian migrations into Turkmenistan and India and a later one with Iberian migrations into South America. There also seems to be a link between Russian horses and the Nokota of North America [115].

While gaited horse breeds mostly retained their popularity in North and South America from the sixteenth century to today (though some breeds such as the Canadian Pacer and Narragansett Pacer went extinct and certain South American breeds such as the Chilean and Costa Rican Saddle Horse seem to have lost the ability for alternative lateral gaits), gaited horse breeds fell out of favor in Europe in the seventeenth century with breeds such as the Bidet Breton, English and French Palfrey, Irish Hobby, Spanish Jennet, Scottish Galloway going extinct and ambling genes being reduced in other gaited breeds such as the Alter Real of Portugal, the Andalusian and Asturcón of Spain, and Mérens of France. Partly this was due to a shift in fashion in favor of larger Arabian, Andalusian, Quarter, and Thoroughbred horses. Secondly, better roadways were developed and carriages became feasible for long-distance transportation, and these were typically pulled by larger coldblooded trotting horses. The same gene (DMRT3) that allows horses to engage in speedy lateral gaits prevents or reduces the quality of the trotting and galloping gaits. Hence the traditional favoring of smaller gaited breeds began to wane. These breeds themselves were also often “improved” by crossing them with blood from other breeds, in part to alter their size or appearance, but also in part to increase their trotting ability. Thirdly galloping horse racing became popular and pacing horses are poor gallopers in general. All of this led to the dilution and decrease in populations of the gaited breeds. Many of the gaited breeds only continued in a feral or semi-feral state, with exceptions to this being the continued favor for gaited horses in the Southern United States, Puerto Rico, South America, Iceland and Faeroe Islands, Wales, Greece, South Africa and West Africa, Turkey, Russia, Central Asia, China, Cheju, Hokkaido, Wushen (Mongolia), and India.

In the late 1800s, however, amblers made a comeback. Bred in the United States in Alabama, Florida, Kentucky, and Missouri, alternatively gaited horses found favor as saddle horses. Subsequently many breeds that were feral were captured and trained so they could become comfortable saddle horses again, as with Galician and Mérens or Europe and the Spanish Mustang and Spanish Colonial Horse of the United States (beginning in 1930s and gaining momentum in the 1990s). In addition, new gaited breeds were developed (Aegidienberger, America Paso Fino, American Gaited Pony, Campolina, Mangolina, Montana Travler, Piquira, National Show Horse, Smokey Valley Horse, Tennuvian, Tiger Horse, Utah Walkalony, Virgina Highlander) or alternative lateral gaits introduced into other breeds, such as the Morgan and Walkaloosa of the United States and the Paso Higueyano of the Dominican Republic, through crosses with gaited breeds. Finally, it has been recognized that the DMRT3 gene is advantageous for harness racing trotters and pacers. Nowadays, ambling horses seem as popular as ever and function as pleasure or trail horses, show horses, or harness racers in many parts of the world.

Figure 2 below (see also Appendix A) shows distribution of alternatively gaited horse breeds along with the alternative lateral gaits with which they are most associated.

## 6. Reasons for the Spread of Alternative Lateral Gaits in Modern Domestic Horses

There seem to be three reasons for the development of alternative lateral gaits in modern horses: adaptive, military, and human comfort.

There is a definite association of the alternative lateral gaits of rack, broken pace, or hard pace, with horse breeds inhabiting mountainous or hilly regions such as the Altai, Ural, Caucasus, Himalayas, Pyrenees, Alps, or Andes ranges, the country of Iceland, and the state of Kentucky in the United States. Such gaits allow for the retention of one foot on the ground at all times (with the exception of the hard pace) and tripedal support phases at intermediate speeds and therefore appear to help horses be more sure-footed and better able to navigate sloped or uneven terrain [194,195,196]. For these same reasons, such gaits would have helped to minimize ground contact forces and prevent joint injury over travel on hard surfaces, such as in rocky mountainous terrain or frozen steppes [4]. Alternative lateral gaits are often also narrow gauge and may allow for enhanced navigation through tight quarters. Racking and pacing gaits are often accompanied by high front leg elevation and knee flexion. Such high-stepping gaits may have allowed for clearance over objects in the horse’s path, such as rocks, stumps, or branches, though an alternative hypothesis is that the elevation of the front legs is merely to reduce length of swinging pendulum for increased speed [46,47] It is also the case that the hard pace, common in horse breeds of Southern Asia, is a very fast intermediate gait, faster indeed than the trot, and this may have had something to do with its origin. It is not clear, however, if this adaptability to mountainous terrain is the reason alternative lateral gaits evolved in the first place, or whether such gaits were artificially selected for in mountainous tribes, probably a bit of both.

Meanwhile, the broken trot commonly occurs in horses living in deserts, marshes, dry grasslands, or tropical forests, including the Akhal-Teke of the Karakum Desert in Turmenistan; the Kathiawari, Marwari, and Sindhi of the Indian Thar Desert; the Nokota of the arid Badlands of South Dakota in the United States; the Nordestino of the arid Caatinga in Brazil; the Campeiro, Mangalarga Marchador, and Pampa of the Araucária Moist Forests of Brazil; and the Buryat, Transkbaila, and Yakut of the Siberian Tiaga. The broken trot thus seems well suited for travel on soft deformable terrain due to periods of tripedal or even quadrupedal support in order helping the horse to not get stuck or slip and fall. The broken trot is also a particularly efficient gait for travel, as we have seen (rivaling the running walk and trot), and so perhaps very useful for covering long distances over somewhat uneven and unlevel arid terrain.

In terms of human breeding or artificial selection of horses, there is a lot of evidence that alternatively gaited horses were favored militarily. Part of this was economical as pacing horses are often small and so could travel just as fast as larger trotting horses but consumed less feed. Yet, part of this seems that warriors on horses exhibiting alternate lateral gaits would not bounce up and down as much and hence could better aim lances, spears, arrows (with hands free), or sword strikes.

Horses with alternative lateral gaits (in particular the hard pace or broken pace) were commonly used by armies in Central Asia, West Asia, Southern Europe, and Southern Asia including the Scythians (ca. 800 BC), Celts (ca. 800–300 BC), Archaemenid Persians (550–339 BC), Macedonians (ca. 339–323 BC), Bactrians (ca. 104 BC), the Chinese Han Dynasty (ca. 100 BC), Sassanid Persians (ca. 700–900 AD), Seljuk Turks (ca. 1037–1308), and Mongols (ca. 1206–1294). Such horses were likely used to pull light war chariots, as well as to hold mounted archers and lancers [176,179,197,198]. Scottish soldiers used Hobby Horses (hobynis) brought from Ireland for military campaigns under Edward I (1296) and this continued for some time, for example, as noted by the poem *The Brus* 112 and 115 (1372) written by John Barbour [199]. The Mamluk treatise *Nihāyat al-su’l wa-al-umnīyah fī ta‘allum a‘māl al-furūsīyah* (1371) depicts Egyptian warriors using spears or lances on gaited horses [151,200]. Fiore dei Liberi’s *Flos duellatorum* (ca. 1410) shows knights jousting or training in sword techniques on ambling horses (indeed in what is perhaps a running walk gait). A painting from Ming Dynasty of China (ca. 1500) shows an archer riding a horse in a broken pacing gait [[117] p. 13]. Thomas Blundeville, in his *The Art of Riding* (1558), notes that Irishmen used ambling Hobby Horses in battle when they fired off darts or threw spears. Finally, Claudio Corte’s work *Il Cavallarizzo* (1562) notes that Spaniards used jennet horses (ginecti) for light cavalry units. So, lateral gaits seem to have had some utility in battle, though again many illustrations and textual descriptions of battles or jousts depict horses at a galloping gait [117] (p. 197). Indeed, recently the Boer cavalry featuring the gaited Cape Boer horse was prominent in the Boer Wars in South Africa (1880–1902) [58]. Such gaited horses were also probably prized for hunting on horseback as with the Galloway horses in the Scottish Taymouth Hours depictions.

Finally, horses with alternative lateral gaits are very comfortable to ride (especially in the rack and running walk) and became a favored mode of transportation with the nobility, as depicted in medieval art. Alternatively gaited horses, lacking periods where all four feet are in the air at once, have less up-and-down motion in the saddle and do not jolt the rider’s back as much as the trot. Because intermediate speed ambling gaits retain ground contact with at least one foot, and sometimes three feet at once, they are also quite secure for horse and rider. Hence, alternatively gaited horses (the *palfredus*) were popular and fetched a high price in the Middle Ages and Renaissance as shown in documents from Christ Church, Canterbury from 1336–1525, and the rolls of Durham abbey from 1456–1457 [201,202,203] (n. 600).

## 7. Genetics of Alternative Lateral Horse Gaits

A breakthrough regarding the genetics of alternative lateral gaits occurred in 2012 [204] when it was discovered that nearly all gaited horses possessed a variant allele (A) as opposed to the wild-type allele (C) of the DMRT3 gene of chromosome 23 found in non-gaited horses. The DMRT3 gene was labeled the “gait-keeper” gene as it controls the number of gaits a horse can (naturally) employ and the transitional speeds between them. Horses without the A-allele tend to have just three gaits–walk, trot, and canter/gallop–whereas horses with the A-allele (especially those homozygous for it) can employ alternative lateral gaits such as the running walk, rack, broken pace, or hard pace and maintain these gaits at intermediate to fast speeds. That is to say, the “gait-keeping” A allele allows the horse to extend lateral-sequence gaits (as with the walk) into intermediate speeds rather than shifting to the diagonal-sequence trot at around 1.5–2.2 m/s, or the asymmetrical gallop at around 4.5 to 6.0 m/s [195].

The wild-type DMRT3 C-allele codes for a protein transcription factor responsible for producing regular bursting patterns in the dI6 interneurons of mammalian spines and thereby coordinating diagonal and contralateral limb movements. The A-allele developed through a single-nucleotide polymorphism (SNP) that introduced a premature stop codon into the gene. The resulting truncated protein transcription factor (possessing only 300 out of 474 amino acids) lacks the ability to produce regular interneuronal bursts in the spine and thereby induce horses to transition from a slow walk to an intermediate speed diagonally coordinated trot, and finally to an asymmetrical contralaterally-coordinated canter/gallop at fast speeds. Instead, horses with the mutated A-allele tend to transition from a slow walk to an intermediate speed laterally-coordinated running walk, rack, broken pace, or hard pace [205,206]. In other words, horse breeds that possess the A-allele (such as the Icelandic, Mangalarga Marchador, Paso Fino, Tennessee Walking Horse, and Saddlebred), when they wish to travel faster, tend to employ alternative lateral gaits based upon the same lateral sequence footfall pattern found in the walk (LH, LF, RH, RF) rather than transitioning to a diagonal trot or asymmetrical gallop. In fact, such horses (especially those homozygous for the A-allele) not only display an unwillingness to engage in trots and gallops but show poor quality versions for beat clarity and speed capacity (as rated by certified judges during breed-specific field tests) of trots and gallops when they do perform them [204,207,208].

Further investigation has found that nearly all alternatively gaited horse breeds, whether located in Europe, Asia, North America, or South America possess a high percentage of the A-allele (>15%) of the DMRT3 “gait-keeper” gene, though there are still a few breeds left to be genotyped, and in several cases only small sample sizes exist ([46,54,65,66,81,85,87,89,92,95,108,109,112,114,209,210,211,212]; see Appendix A for additional sources). Only a few inconsistencies still exist, such as with the occasionally gaited Dongola of West Africa and Nokota of North America, who, so far, have not been found to possess the A-allele of the DMRT3 gene [210,211]. Larger sample sizes might eventually change this. Importantly, the broken trot gait may not be linked at all or only in a minor way to the A-allele of the DMRT3.

Effects peculiar to the different alleles of the DMRT3 gene have also been observed. For example, it has been found that in Icelandic Horses pacing generally requires a genotype homozygous for the DMRT3 A-allele (i.e., AA), whereas horses with a heterozygous genotype (CA) are typically only able to perform the rack (tölt). More particularly, 94% of five-gaited Icelandic Horses who could perform both the tölt and the flying pace were homozygous for the A-allele (i.e., AA), whereas while 88% of four-gaited Icelandic horses who could only perform the tölt but not the flying pace had at least one copy of the A-allele (about two-thirds having the CA genotype and one-third the AA genotype). And only 29% of Nordic Trotters with the CA DMRT3 genotype could perform a quality hard pace [208,209,213,214]. More study, however, needs to be performed to see if this holds true for other pacing horse breeds as well. Fonseca et al. [215] found that Mangalarga Marchador horses of the homozygous AA DMRT3 genotype had greater diagonal advanced placements (AA = 31.7, CA = 28.9) and smaller periods of diagonal bipedal support (AA = 35.5, CA = 41.0) in the broken pace (marcha picada) than those that were of the heterozygous CA genotype. More studies of this sort would be informative with other breeds. In addition, while homozygosity for the A-allele reduced the quality of the trot and gallop in Icelandic Horses, having a single copy of the A-allele was beneficial in warm-blooded harness racers, whether they trotted or paced, as it encouraged the horse to sustain the trot or hard pace at higher speeds rather than switching to a gallop [204,216,217,218,219,220].

It also turns out that other genes besides the DMRT3 are involved in the production of alternative lateral gaits, of which we are only beginning to understand. As Petersen et al. [209] point out, breeds with the DMRT3 A-allele on chromosome 23 display various types of alternative lateral gaits, and so “it appears that this locus does not itself explain the entirety of the variation in gait present in domestic horses … (but rather) that gait is a polygenic trait, and … variations among breeds are determined by modifying loci”.

In the first place while homozygosity for the DMRT3 A-allele (i.e., an AA genotype) seems necessary for the ability to pace in Icelandic and Hokkaido horses (only 4–6% of CA genotype Icelandics could pace and 0% of CA Hokkaido horses) and perhaps in other horse breeds as well, it does not seem sufficient. For only 70–94% of Icelandic and 86% of Hokkaido horses homozygous for the A-allele were reported to pace [94,214]. This could merely reflect the fact of upbringing and training (though not maternal example, as Amano et al. [94] noted) or it could be due to further genetic factors.

In the second place, the DMRT3 A-allele and its various genotypes are not great predictors of pacing versus trotting ability in warmblooded or coldblooded harness racers. While 71% of homozygous AA genotype French Trotters were indeed trotters and only 29% pacers, the opposite was the case with Finnhorses where ca. 78% of the AA genotype were pacers and only ca. 22% trotters. Similarly, while 98% of heterozygous CA genotype French Trotters were trotters and 2% pacers, ca. 82% of heterozygous CA genotype Finnhorses were trotters and ca. 18% pacers [221,222]. Moreover, Standardbred horses are nearly all homozygous for the AA allele but ca. 56–66% trot while ca. 34–44% pace [204,208,214,223]. Again, Tennessee Walking Horses (as well as the National Spotted Saddle Horse) are almost all homozygous for the DMRT3 A-allele, and though they are famous for the running walk gait, many members of the breed can perform other alternative lateral gaits such as the rack, broken pace, or broken trot [33,210,211]. Nor did the particular genotype of the DMRT3 gene (AA, CA, or CC) correlate well with whether or not American Saddlebreds were three- or five-gaited (i.e., were shown in slow gait (broken pace) and rack in addition to walk, trot, and gallop). In fact, three- and five-gaited horses had nearly the same proportions of genotype: AA (7%, 3%), CA (26%, 24%), and CC (24%, 26%) [54].

In the third place, the A-allele of the DMRT3 gene appears to have little to no role in the generation of the lateral sequence diagonal-couplet gait (i.e., the broken trot or fox trot). For the DMRT3 AA genotype is nearly fixed (100%) in the racking and pacing Icelandic Horse as well as in the Missouri Fox Trotter [211]. And the CC genotype is nearly fixed (100%) in other breeds that engage in the broken trot including the Akhal-Teke (glide gait), Karbarda (tropota gait), Transbaikal (tropota), Yakut (tropota), and Marwari (revaal gait) [89,96,108,109,175,210,224]. Similarly, Colombian Paso Fino horses that perform a rack (fino classico) have a nearly fixed A-allele with the following genotype frequencies: AA (0.94–1.00), CA (0.00–0.01), and CC (0.00–0.05). However, Colombian Trocha horses that perform a broken trot (trocha) have a nearly fixed C allele, with the following genotypes: AA (0.00–0.03), CA (0.02–0.15), and CC (0.82–0.98)—though the CA genotype might help in producing a more distinct trocha (or trot) gait [56,208,210]. Parallel findings occur with the Brazilian Mangalarga Marchador breed. Mangalara Marchador horses that preferred the broken pace (marcha picada) almost all possessed the A-allele of the DMRT3 gene, having the following genotypes: AA (0.31–0.87), CA (0.13–0.65), CC (0.00–0.05), whereas horses that preferred the broken trot (marcha batida) had the following genotypes: AA (0.00–0.15), CA (0.00–0.34), CC (0.85–0.94) [65,66,225,226,227]. A study of Brazilian Campolina horses, however, found similar gene frequencies in horses that perform the broken pace (marcha picada) or the broken trot (marcha batida). Horses that performed the marcha picada had genotype frequencies of AA (0.12), CA (0.88), and CC (0.00), while horses that performed the marcha batida had genotype frequencies of AA (0.44), CA (0.56), and CC (0.00) [65]. Hence, it seems that while at least one copy of the A-allele is necessary for the performance of the rack and broken pace, and perhaps two copies (i.e., homozygosity) for the performance of the hard pace, the A-allele is not required at all for the performance of the broken trot.

Equine scientists are still untangling the other genes involved in the production of alternative lateral gaits and their variations. Initial explorations have found various candidate genes on chromosomes 1, 19, 23, and 30 that correlate to some degree with the particular alternative lateral gait displayed by a horse, but much more work needs to be performed [22,33,94,98,221,227]. Some genes seem to be associated with the rapidity and length of the horses’ steps, which tend to be quick and collected in Paso Fino horses but slower and more extended in other breeds [56]. There are probably also genes related to the degree to which the front feet are elevated or not. Finally, there seem to be several genes related to speed. In particular, the MSTN “speed gene” on ECA 18, which produces muscular myostatin is of some importance. Horses with the variant C-allele of the MSTN gene (which increased in frequency in horses from 900–1400 AD) are faster that those homozygous for the wild-type T, and in particular, horses that are homozygous for the variant (CC) are capable of short bursts of speed, and horses that are heterozygous (CT) are better at middle distance racing, and individuals without the C-allele (TT) have the greatest endurance. Moreover, horses with the variant C-allele in the CKM gene have greater endurance, horses with the variant T-allele in COX4K2 gene have greater durability, while individuals homozygous for variant G-allele in the PDK4 gene have greater short-distance speed [172,218,228,229,230,231,232,233,234]. However, understanding of the genetic factors undergirding and controlling the various alternative lateral gaits is still in its infancy.

## 8. Conclusions

In addition to the three “natural” gaits of walk, trot, and canter/gallop, several “gaited” horse breeds possess the alternative lateral gaits of running walk, rack, broken pace, broken trot, or hard pace. Such gaits are intermediate speed but, like the walk, involve a sequence of ipsilateral limbs and tend to maintain contact with the ground with at least one limb throughout the duration of the stride (the exception being the hard pace). The gaits of horses can be organized according to the following schema (gait: speed, symmetry, temporal limb sequence, temporal hoof couplets, chronicity, beats, biomechanics).

Natural gaits:

walk: slow, symmetrical, ipsilateral-sequence, singlefoot, even, four-beat, inverted pendulum;

trot: intermediate, symmetrical, diagonal-sequence, diagonal-couplet, even, two-beat, spring mass;

canter: fast, asymmetrical, contralateral-sequence, contralateral and diagonal-couplet, uneven, three-beat, spring mass and collisional;

gallop: fast, asymmetrical, contralateral-sequence, contralateral-couplet, uneven, four-beat (at slower speeds) to two-beat (at very fast speeds), collisional.

Artificial lateral gaits:

running walk: intermediate, symmetrical, ipsilateral-sequence, singlefoot, even, four-beat, inverted pendulum;

rack: intermediate, symmetrical, ipsilateral-sequence, singlefoot, even, four-beat, spring mass;

broken pace: intermediate, symmetrical, ipsilateral-sequence, ipsilateral-couplet, uneven, four-beat, spring mass;

hard pace: intermediate, symmetrical, ipsilateral-sequence, ipsilateral-couplet, even, two-beat, spring mass;

broken trot: intermediate, symmetrical, ipsilateral-sequence, diagonal-couplet, uneven, four-beat, spring mass.

While the trot, hard pace, and gallop are capable of great speed and are quite efficient for long-distance travel, alternative lateral gaits offer “easy” or “smooth” gaits for the rider and provide sure-footedness for the horse on uneven or slippery terrain. This is likely the reason why they have arisen on at least two separate occasions in equids.

While basil perissodactyls likely were only capable of the walk, trot, and gallop, as with tapirs, the racking gait has been definitively identified in some Pliocene tridactyl equids of Laetoli, Tanzania traveling across slippery ash deposits, and perhaps in some Miocene tridactyl equids of Barstow, California. A plausible hypothesis that accords with ichnological and paleontological evidence is that alternative lateral gaits first arose in the early Miocene, either in tridactyl stem-equine anchitheriines such as *Parahippus* that had long legs with a large hooved central digit allowing for an unguligrade posture and a spring foot or in tridactyl Equinae such as *Merychippus*, and continued on in the early Miocene (mostly) monodactyl Equini and tridactyl Hipparionini tribes, as well as in the Old World hipparionins through the Pliocene, but were lost in the lineage of monodactyl Equini of the late Miocene (i.e., *Dinohippus*) that led to the emergence of the modern horse genus *Equus*.

Alternative lateral gaits evolved in horses for a second time soon after or just before horses were domesticated around two to three thousand years ago in Central Asia. This has been traced to the SNP that yielded the A-allele of the DMRT3 “gait keeper” gene which caused horses to employ intermediate speed lateral gaits instead of the trot. Such gaited horses, usually with a pace or broken pace, were directly or indirectly spread through the Scythians to Southern Asia, Western Asia, and Eastern Europe. Later the medieval Sassanid Persians and Turks also favored horses with a pace or broken pace, which is why such gaited breeds are common today in Turkmenistan, Tajikistan, Uzbekistan, Azerbaijan, Turkey, and Iran. Such gaits, with continual unipedal ground contact even at intermediate speeds, seemed useful for military and transportation purposes on mountainous terrain. The Celts, who may have imported Scythian horses in the fifth or fourth century B.C., developed their own breeds of gaited horses, often with a rack or running walk, and these were spread by them into Spain and Britain. We find such gaited horses becoming very popular in Europe in the Middle Ages as riding horses of the nobility, likely due to their comfort since the lack periods where all four-limbs are off the ground. There also seems to have been a secondary use of gaited breeds as hunting or military horses, such as with the Hobby Horse of Ireland. Around the same time the racking horse was introduced into Iceland. Later on the Spaniards spread such gaited horses into the New World.

Another alternative lateral gait, one that does not seem to be controlled by the DMRT3 gene, but which is very efficient, is that of the broken trot. Such a gait occurs in breeds occupying desert or marshy regions of Siberia, India, and Turkmenistan, as well as in North American horses, including the famous Missouri Fox Trotter and Carolina Marsh Tacky, and some breeds of Brazil and Columbia.

Such gaited horses—prized for military use and transportation by ancient Scythians, Celts, Persians, Greeks, Chinese, Turks, Mongols, and Spaniards—continue to be of great service and enjoyment to humans today. While Thoroughbreds and Quarter Horses, along with Arabians and Andalusians, are common breeds, horse farms breeding and training alternatively gaited horses have become numerous throughout the United States and world. For alternatively gaited horses are popular as pleasure or road animals due to their smooth and comfortable intermediate speed gaits, as well as in the show arena. Harness racing is also a popular equestrian sport with both trotting and pacing horses. For all of these reasons alternative lateral gaits and gaited breeds should continue to rise in popularity among horses of the future (unless this is just the wishful thinking of the author).

## Figures and Tables

**Figure 1 animals-13-02557-f001:**
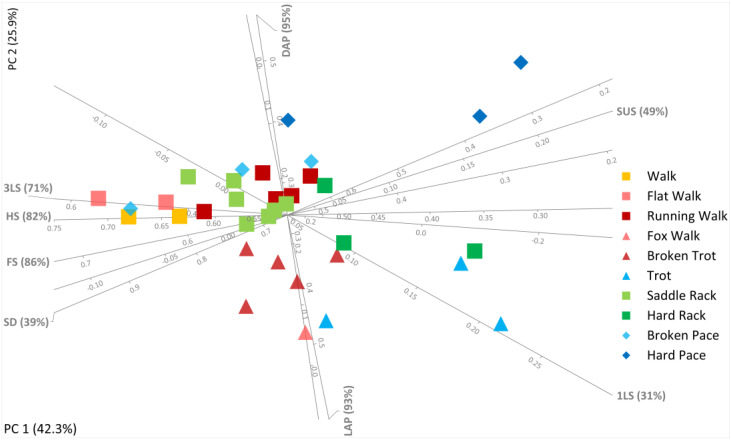
Modified Hildebrand diagram for symmetrical horse gaits, plotting lateral advanced placement [LAP] (limb phasing value or % of stride duration separating ipsilateral foot contacts) vs. hind foot stance duration [HS] (duty factor or % of stride duration hind feet are on the ground), as well as stride duration [SD] (seconds), front foot stance duration [FS], seconds, percentage of tripedal support throughout stride duration [3LS], percentage of unipedal support throughout stride duration [1LS], and percentage of stride duration spent in suspended phases where all four feet are off the ground together [SUS]. Values following these abbreviations show what percentage of principal component is based on the given parameter. Square gaits (walk, flat walk, running walk, saddle rack, hard rack) are symbolized by a square, diagonal couplet gaits (fox walk, broken trot (i.e., fox trot), and trot) are symbolized by a diamond, and lateral couplet gaits (broken pace (i.e., stepping pace) and hard pace) are symbolized by a triangle. Square gaits tend to have longer stride durations (and duty factors), with diagonal gaits intermediate in value, and lateral gaits (and asymmetrical gaits) with the least stride durations. Diagonal gaits have the largest lateral advanced placements (limb phasing values) while lateral gaits have the smallest. Data points are averaged values from the following sources [2,15,23,30,42,43,50].

**Figure 2 animals-13-02557-f002:**
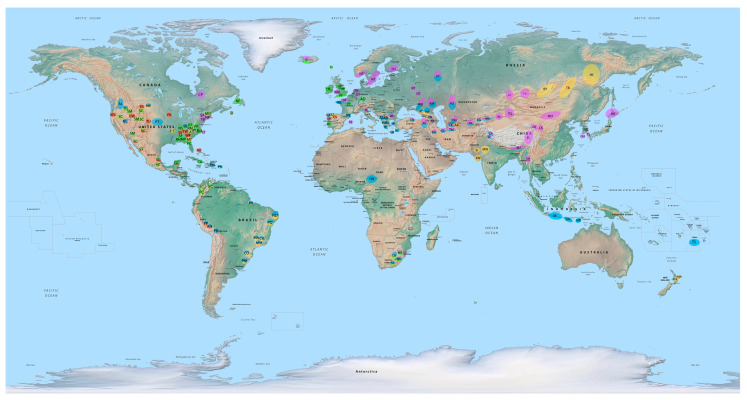
Map showing world-wide distribution of horse breeds with alternative lateral gaits. Gaits are color-coded with yellow indicating the broken trot (fox trot), red the running walk, green the rack (tölt), blue the broken pace (stepping pace), and purple the hard pace. When horses perform more than one gait the circle surrounding their abbreviation is multi-colored. The following are the horse breeds denoted and their abbreviations: AB = Abaco Barb, AD = Aegidienberger, AK = Akhal-Teke, AL = Albanian, AN = Andean, AP = Appaloosa, AR = Arravani, AS = Appalachian Singlefoot, AT = Altai, AY = Andolu Yerli, AZ = Azteca, BB = Bidet Breton (extinct), BH = Banker Horse, BS = Basuto, BT = Bulgarian Trotter, BU = Bhutia, BY = Buryat, CA = Campolina, CB = Cape Boer, CH = Cheju, CI = Caspian, CK = Chakouyi, CO = Campeiro, CN = Canik, CP = Canadian Pacer (extinct), CR = Cretan, CS = Castilian (extinct) (same location for SJ = Spanish Jennet (extinct)), CT = Chincoteague Pony, CU = Cuban Paso, DH = Dutch Harness Horse, DN = Dongola, DT = Datong, KC = Karachai, KD = Kabarda, FC = Florida Cracker, FG = Ferghana (extinct), FH = Finnhorse, FI = Faeroe Island Pony, FR = French Trotter, FT = Missouri Fox Trotter, GA = Galician, GC = Galiceño, GR = Garrano, GT = German Trotter, GW = Galloway Pony (extinct), HB = Hobby Pony (extinct), HK = Hokkaido, IC = Icelandic, IT = Italian Trotter, JV = Java (extinct), KA= Kaimanawa, KD = Kabarda, KG = Kyrgyz, KK = Karabakh, KM = Kalmyk, KR = Karabair, KS = Kentucky Mountain Saddle Horse (same location for KN = Kentucky Natural Gaited Horse and for VS = Smokey Valley Horse), KU = Kurdi, KW = Kathiawari, KZ = Kazakh (same location for KS = Kushum), MA = Mangolina, ME = Mérens, MI = Mytilene, MJ = Marajoara Island Horse, MO = Gaited Morgan, MM = Mangalarga Marchador, MP = McCurdy Plantation, MR = Timor, MT = Carolina Marsh Tacky, MW = Marwari, NA = Nati, NC = North American Curly, ND = Nordestino, NF = Newfoundland Pony, NG = Nooitgedacht, NK = Nokota, NP = Narragansett Pacer (extinct), NS = National Show Horse, NT = Nordic Trotter, OT = Orlov Trotter, PA = Canadian Pony of the Americas, PC = Colombian Paso Fino, PE = Peneia, PF = Palfrey (extinct), PH = Paso Higueyano, PI = Pindos, PM = Pampa, PQ = Piquira, PP = Peruvian Paso, PR = Puerto Rican Paso Fino (same location for RC = Puerto Rican Criollo), PT = Paint, RE = Alter Real, RH = Racking Horse, RM = Rocky Mountain Horse (same location for OK = Mountain Pleasure Horse), RO = Rhodian, RT = Russian Trotter, SA = Standardbred (Pacer), SB = American Saddlebred, SC = Spanish Colonial, SH = Shan, SI = Sindhi, SF = North American Singlefooting, SI = Sable Island, SL = Afrikan Saalperd, SM = Spanish Mustang, SP = Spiti, SS = National Spotted Saddle Horse, ST = Sierra Tarahumara, SW = Sandalwood, TA = Taishuh, TB = Trottingbred, TC = Colombian Trocha Pura (same location for TG = Colombian Trocha y Galope), TG = Tiger Horse, TH = Thessalian, TI = Tibetan, TK = Turkmene (extinct), TL = Transbaikal, TM = Turkoman (same location for NI = Nisean (extinct)), TN = Tushin, TR = Montana Travler, TS = Tongan Singlefooter, TU = Tennuvian, TV = Tuva, TW = Tennessee Walking, UW = Utah Walkalony, VC = Venezuelan Criollo, VH = Smokey Valley Horse, VY = Vyatka, WB = West Black Sea, WL = Walkaloosa, WM = Welsh Mountain Pony, WP = American Walking Pony, WU = Mongolian Wushen, YK = Yakut, YL = Yili, YM = Yamud, YQ = Yanqi, ZA = Zaniskari. Data from personal observations and following sources [16,20,21,22,35,36].

## Data Availability

All data for this review article taken from references.

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
