# Peer review of "The Characteristics, Distribution, Function, and Origin of Alternative Lateral Horse Gaits"

_animals, 2023, doi:10.3390/ani13162557_

Round 1
Reviewer 1 Report
Line 68-77: improve the format of the paragraph, better division between gaits described would help understanding and reading.
Line 214: typo, correct hose to horse
Line 215: possibly typo mistake, head vertical displacement in mm?
none
Author Response
RESPONSE TO REVIEWER 1 COMMENTS:
Point 1: Line 68-77: improve the format of the paragraph, better division between gaits described would help understanding and reading.
Response 1: Paragraph reformatted to separate gaits described and also brief legend provided
Point 2: Line 214: typo, correct hose to horse
Response 2: Corrected to horse
Point 3: Line 215: possibly typo mistake, head vertical displacement in mm?
Response 3: Units corrected to cm (good catch)
Reviewer 2 Report
Dear Author,
I congratulate you for your work, because it is recognized that you did a deep study, but an Article (even a review), with close to 15 thousand words and 405 references cannot happen... a book yes, but not a Review Article. It should be, as the name implies, a review and not include everything that appears in the literature on the subject. Need to reduce a lot, follow some suggestions.
In general, the work should focus on the movements that lateralize and not refer to the others, reorganize the work: History (reduced), changes, genes and sensors through new methods in the evaluation of biomechanics. It cannot tell everything that exists in the literature, but a review that allows readers to find reduced information on the main studies on the subject.
Line 36 - Gallop and Canter are not the same thing, so you can't say "or" but Gallop and Canter.
Does it take 7 references for a line of sentence, no, in this case 2 at the most... and as this example has many more below, I will mention some, but there is more.
Line 39 and 40 - At no time does the "Walk" progress of the horses have a moment of suspension, an error considered serious, and even indicates 10 a reference to this. This sentence is not true "...or period where all four limbs are off the ground at once". The trot and gallop have a suspension moment, the walk does not!
One way to reduce the size of the study is to focus only on the gaits it presents, so that these errors do not arise.
Line 43 - Put 14 references for a sentence (it cannot happen and as this one has dozens of this type throughout the article), reduce to 2 to 3 in each place, in some a reference.
The natural Gaits of horses are, Walk, Trot, Gallop and Retreat (these are the natural gait, according to the "International Equestrian Federation") so, focus on what it presents, there are breeds of horses that have Gaits that for them are today in natural days and are not part of this 4, but those that the author presents.
Line 68 to 77 - where does this information come from?
Line 108 - what do you mean by "(and sometimes canter)" ?
You don't need to describe Step, Trot and Gallop, describe the gaits you refer to, focus on the theme of the Review.
Line 131 - Why does it display like this: "0.55-0.30" and not 0.30-0.55? Correct and there are several throughout the work to correct (I only indicate this one but correct all that need to be corrected).
Gallop to clarify, Normally 3 beats, in running gallop 4 beats, in some dressage exercises 4 beats...
Where is the publication that says that normally there are 4 beats?
Starting at Line 172 - At this point (2.1) he talks about gaits then he talks about the probable origin of some horses... this extension is difficult to read. Don't get lost in too much history.
Line 339 - Attention to certain assumptions, there should be no assumptions in science.
Line 350 and 351 - the difference is minimal and in the last value it is even higher, so this sentence has to be corrected "that mean lactate blood concentrations were generally higher in the tölt" because it does not correspond to what it presents, the values presented are identical.
At some point in the article, a bias favoring some Gaits is visible, pay attention to that.
Line 364 - Put 17 references, why?
Line 372 - For 3 lines you present 20 references, it doesn't make sense.
Line 443 - There are many horse breeds in Brazil that are not included in this type of gait, so the normally that follows makes no sense
And the references to horses in Brazil, there are several studies, and with Mangalarga there are two breeds, there is the Mangalarga Marchador and just Mangalarga (which sometimes in the latter add Mangalarga Paulista), only these two breeds have well over half a million registered horse. Compared to other breeds, the number of animals is much higher, many studies.
More texts and references in this range of lines, and without the ability to be a review, remembering, a review article is to reduce and discard the most important.
Line 1114 - How do you know that the Gaits are of good or bad quality, did the cited authors quantify?
Line 1121 - 29 References ?
Line 1205 - There are already genomic studies that indicate how to evaluate the fastest or most resistant horses.
Line 1247 - There is no suspension on the walk!
Conclusion - It is too long, you need to reduce it a lot and pay attention to what a conclusion is. Do not post dates or results. Just indicate that XXX evolution provided animals for XXX and with capabilities of xxx.
I only suggested a few points, since the others are identical corrections.
Author Response
Point 1: I congratulate you for your work, because it is recognized that you did a deep study, but an Article (even a review), with close to 15 thousand words and 405 references cannot happen... a book yes, but not a Review Article. It should be, as the name implies, a review and not include everything that appears in the literature on the subject. Need to reduce a lot, follow some suggestions.
Response 1: Thank you. I have reorganized and combined some sections to reduce overall length and avoid repetitions. I have also omitted all non-essential references.
Point 2: In general, the work should focus on the movements that lateralize and not refer to the others, reorganize the work: History (reduced), changes, genes and sensors through new methods in the evaluation of biomechanics. It cannot tell everything that exists in the literature, but a review that allows readers to find reduced information on the main studies on the subject.
Response 2: I have now moved discussion of natural gaits to introduction only so focus later on is on alternative lateral gaits. As noted above I have reduced historical discussion by combining sections and same for biomechanics. I have tried to get key information to readers now without discussing everything known.
Point 3: Line 36 - Gallop and Canter are not the same thing, so you can't say "or" but Gallop and Canter.
Response 3: Changed to canter/gallop or described the gaits separately.
Point 4: Does it take 7 references for a line of sentence, no, in this case 2 at the most... and as this example has many more below, I will mention some, but there is more.
Response 4: Throughout I have reduced number of references, only giving key ones.
Point 5: Line 39 and 40 - At no time does the "Walk" progress of the horses have a moment of suspension, an error considered serious, and even indicates 10 a reference to this. This sentence is not true "...or period where all four limbs are off the ground at once". The trot and gallop have a suspension moment, the walk does not! One way to reduce the size of the study is to focus only on the gaits it presents, so that these errors do not arise.
Response 5: There may be some misunderstanding here. I tried to express the fact that there was no suspended phase at all in the walk (in agreement with reviewer and not that there were suspended phases which would have been an error) but may not have expressed it clearly enough. So I have rewritten the sentence to talk about continual ground contact.
Point 6: Line 43 - Put 14 references for a sentence (it cannot happen and as this one has dozens of this type throughout the article), reduce [references] to 2 to 3 in each place, in some a reference.
Response 6: Reduced references.
Point 7: The natural Gaits of horses are, Walk, Trot, Gallop and Retreat (these are the natural gait, according to the "International Equestrian Federation") so, focus on what it presents, there are breeds of horses that have Gaits that for them are today in natural days and are not part of this 4, but those that the author presents.
Response 7: I have minimized discussion of natural gaits to focus on presentation of alternative lateral gaits as reviewer suggests (moving all discussion to introduction only).
Point 8: Line 68 to 77 - where does this information come from?
Response 8: This is my own schema so I did not list any references; but the basic information comes from the previously cited references.
Point 9: Line 108 - what do you mean by "(and sometimes canter)"?
Response 9: Changed to canter/gallop to reflect that these are two different but related gaits. There is some discussion as to whether all horse breeds can canter, or canter without training, but it is best not to get into this here, so ‘sometimes’ omitted.
Point 10: You don't need to describe Step, Trot and Gallop, describe the gaits you refer to, focus on the theme of the Review.
Response 10: Agreed. Done.
Point 11: Line 131 - Why does it display like this: "0.55-0.30" and not 0.30-0.55? Correct and there are several throughout the work to correct (I only indicate this one but correct all that need to be corrected).
Response 11: The numbers are presented in terms of values from slower to faster speeds, so sometimes the value will increase, and at other times decrease. I have now indicated this in text when values proceed from higher to lower as speed of gait increases.
Point 12. Gallop to clarify, Normally 3 beats, in running gallop 4 beats, in some dressage exercises 4 beats... Where is the publication that says that normally there are 4 beats?.
Response 12. This is where there is some debate on terminology. I have used canter for three-beat version of asymmetrical gait and then gallop for four-beat version and given references for authors giving this distinction. I have now also given reference for gallop here. I have also clarified beats in gallop as in some cases it can sound like just two very fast beats, i.e. thundering hooves, as another reviewer noted.
Point 13. Starting at Line 172 - At this point (2.1) he talks about gaits then he talks about the probable origin of some horses... this extension is difficult to read. Don't get lost in too much history.
Response 13. I have now removed historical discussions from presentation of gaited horse breeds. Where necessary historical material transferred to discussion of origin of alternative lateral gaits.
Point 14. Line 339 - Attention to certain assumptions, there should be no assumptions in science.
Response 14. I have now moved biomechanical information into this section to support assertions with references and removed suppositional claims.
Point 15. Line 350 and 351 - the difference is minimal and in the last value it is even higher, so this sentence has to be corrected "that mean lactate blood concentrations were generally higher in the tölt" because it does not correspond to what it presents, the values presented are identical.
Response 15. Yes, thanks. I clarified the data a bit. Stefánsdóttir [110, their table 1] reports mean lactate blood concentrations in the tölt (1.07, 1.48, 4.66 mmol/l) versus the trot (0.92, 1.27, 4.92 mmol/l), and yes the last value is higher in trot than the tölt. They did find the first two values higher in the tölt than trot with statistical significance. But again as you note the last value was higher in trot. I have clarified this and noted at highest speed of 5.5 m/s value was higher in trot than tölt and what this means. So I have omitted word ‘generally’ to be more specific here.
Point 16. At some point in the article, a bias favoring some Gaits is visible, pay attention to that.
Response 16. Yes, I have now tried to support claims with biomechanical information and references; admittedly discussion of which gaits are most comfortable is somewhat subjective.
Point 17. Line 364 - Put 17 references, why? Line 372 - For 3 lines you present 20 references, it doesn't make sense. Line 1121 - 29 References? More texts and references in this range of lines, and without the ability to be a review, remembering, a review article is to reduce and discard the most important.
Response 17. References reduced to key ones throughout.
Point 18. Line 443 [sic 433] - There are many horse breeds in Brazil that are not included in this type of gait, so the normally that follows makes no sense.
Response 18. Good point. I have changed this from ‘Brazilian horses typically perform a broken pace’ to ‘Several Brazilian horse breeds’. I meant to say gaited Brazilian horses but the previous sentence works even better.
Point 19. And the references to horses in Brazil, there are several studies, and with Mangalarga there are two breeds, there is the Mangalarga Marchador and just Mangalarga (which sometimes in the latter add Mangalarga Paulista), only these two breeds have well over half a million registered horse. Compared to other breeds, the number of animals is much higher, many studies.
Response 19. Added Mangalarga Paulista here (which I do reference later). I have just one reference here as it is specifically in reference to newer marcha de centro gait; other references follow later in discussing other gaits such as marcha picada.
Point 20. Line 1114 - How do you know that the Gaits are of good or bad quality, did the cited authors quantify?
Response 20. Noted what characteristics of gait rated, i.e. beat clarity and speed capacity, and that rating was performed by certified judges during breed-specific field tests.
Point 21. Line 1205 - There are already genomic studies that indicate how to evaluate the fastest or most resistant horses.
Response 21. Added more references here.
Point 22. Line 1247 - There is no suspension on the walk!
Response 22. Yes, added walk to list of gaits which have no periods of suspension.
Point 23. Conclusion - It is too long, you need to reduce it a lot and pay attention to what a conclusion is. Do not post dates or results. Just indicate that XXX evolution provided animals for XXX and with capabilities of xxx.
Response 23. Conclusion reduced and redone with dates and results omitted.
Reviewer 3 Report
The author did an enormous job describing the different Alternative horse gaits, in locomotor and phylogenetic terms. The manuscript is very well organized and easy to read, it contains a lot of useful information.
Strengths: The topic is very interesting, and the approach is very new.
Weaknesses: It is partly repetitive. It could be shortened or it could deepen some aspects such as locomotion
The term nature is redundant with the term origin. either try to define it, or remove it where it matches the term origin even in the tittle.
A dimension of the number of animals that each breed has, or at least the most important breeds, would help to measure reality.
I think there are ideas that are repeated unnecessarily, and in general the article could be shorter. Eg
How horse gain, lost and regain the ability to perform alternative lateral gaits is commented several times.
In the bibliography, in book citations, please detail the chapter
Author Response
Reviewer # 3:
Point 1: It is partly repetitive. It could be shortened or it could deepen some aspects such as locomotion.
Response 1: I have shortened the paper and combined some sections to avoid repetition (such as art and history; gait descriptions and biomechanics) and added a bit more on association of gaits with terrain.
Point 2. The term nature is redundant with the term origin. either try to define it, or remove it where it matches the term origin even in the tittle.
Response 2: I used term characteristics instead of nature throughout.
Point 3. A dimension of the number of animals that each breed has, or at least the most important breeds, would help to measure reality.
Response 3: Very good idea. I included new column giving number of individuals in each breed in the supplementary material section.
Point 4. I think there are ideas that are repeated unnecessarily, and in general the article could be shorter. Eg How horse gain, lost and regain the ability to perform alternative lateral gaits is commented several times.
Response 4: I have shortened the paper and combined some sections to avoid repetitions (see also response 1).
Point 5. In the bibliography, in book citations, please detail the chapter.
Response 5: I have given chapter titles and pagination where appropriate in bibliography now.
Reviewer 4 Report
Most of my review contains just informations for the author. The 2 first paragraphs are also for the editor.
The article is intended to give a comprehensive overview on the gaits of horses. This aim is reached, although the greatest stress is placed on the “alternative gaits”. Working since years on horses, the reviewer congratulates the author for his impressive compilation which includes as many as 405 citations. In spite of the great number of my remarks (distributed on nearly 30 pages!), all of them are “minor corrections”. As soon as these are made, I strongly recommend to accept and publish this paper.
The text is organised according to the gaits: running walk, rack or tölt, roken pace, (hard) pace, stepping pace or amble, broken trot. All of them occur rarely among mammals. There are, however, some items I would like to have reconsidered and – perhaps – improved:
Abstract, line 1: “Origination” should be replaced by the shorter and clearer “origin”
Introduction, Lines 44 and 45: “...gallop (or canter)...” looks like addition of a synonym. This is o.k., but on p. 2, 1st line, the reader finds: “..the four-beat gallop (or three-beat canter) is a ...fast gait....”. This is confusing, because the so-called four-beat gallop occurs only at slow speeds, sometimes intended by the rider, sometimes not. Three beats are typical for the fastest among gaits, only at very high speeds (>15m/s, “Renngalopp” in German, in English “gallop” in contrast to the slower, three-beat “canter”) just 2 double beats can be heared, which can be interpreted as “four beats”.
Lines 60, 62, 64 “...four limb suspension” is difficult to understand. “Suspension” means without ground contact. This is clear, but why the addition of “four limbs”.
P. 92 ff, 106 ff: The taxonomy of wild equids in this article is unusual, since Asian wild asses are lumped with African wild asses as E. (asinus), sometimes E. hemionus. All wild-living equids are confined to using the classical gaits: walk, trot, gallop.
Chapter 2, Alternative gaits.
Line 108; “.. gallop (and sometimes canter)...” implies a difference between the two gaits, which does not exist. If the author just deletes “and sometimes”. this problem is solved. I personally prefer to calling “canter” the three-beat locomotion which can be maintained over long distances, and “gallop” the four-beat locomotion used in racing over short distances of 2400 m or less (see above, the English meaning)
Lines 133-134 I do not know anything about a “park trot” of Morgan horses, and the sentence may therefore stay as before. But I am definitely missing the “passage”, a sustained trot, which is performed by all dressage horses.
Lines 140-141 This can be understood as a precise description of the “gallop” – as noted above, on lines 44 f. I just do not comprehend, why this is summarised under the headline “alternative lateral horse gaits.”
Fig. 1: The caption should explain what the numbers at the far ends of the rays do mean.
Lines 180 ff. the author indulges in mechanical ideas concerning the propulsion of the horses. This must be taken with care, because no really convining evidence is presented.
Line 187 f.: The “rear limbs will have a low, sweeping stride”, but the “frontlimbs a short elevated stride...” This seems to ignore the simple fact, that stride length in both limbs must necessarily be the same (unless the animal becomes longer and longer or shorter and shorter with each step.
Line 192: I am not entirely happy with the repeated note that striding follows the laws of the “inverted pendulum”. This is true for the stance phases, but self-evident for any locomotion on limbs. The swing phases follow the laws of a suspended, hanging pendulum, and this explains in many cases why the limbs are elevated (to shorten the limb pendulum and therefore to increase the speed of the foreswing), not in order to avoid obstacles – as the author says repeatedly.
Line 220: “Heels” should be placed in quotation marks, to emphasise that the “heels of the hove” are mentioned in the meaning of blacksmiths.
From line 223 on, the author goes into detail regarding the alternative gaits and their origins and occurrence.
Line 510 ff. – 583 deals with “hard pace, or flying pace”, a gait I would consider (in another context, however) together with trot.
Chapter 3.1 ,
Line 565 – 759 are the most elaborate recordings of historical literature which is known to me! I can only repeat what he says. The citations are more exact and detailed than anything I have seen before!
Line 864: Wouldn´t be “grassy” better than “glassy”?
Lines 908 – 921 contain estimated absolute ages. These assumptions in my opinion are risky and perhaps not correct. They do not become more reliable if repeated on lines 948 and 949.
Lines 956-974 review early domestication of horses, and consider the allele A of DTRM3 gene, which seems to characterise alternative gaits since nearly 10.000 years ago. An interesting, and not usually recognized item is that Philip of Macedonia imported in the year 339 BC Ferghana horses, earlier than the famous Chinese expeditions. In addition the Scythian culture in the east, as well as the Celts in the west are connected with gaited horses. What the author says, deviates from common views, but sound very plausible.
Understanding of the author is not improved by his using uncomman abbreviations (CE,
BCE)
Lines 1072 – 1087 draw a connection between the alternative gaits and the terrain on which the animals live. This part is repeated below in Chapter 6, and I do not see a reason why this aspect is dealt with at two places.
Lines 1088 – 1217 go into detail concerning the frequency of the A-allele of DTRM3-gene –as well as other genes! - in breeds of gaited horses. I have not controlled, whether there is any repetition or contradiction to the above-noted (on lines 956-974).
Chapter 6, lines 1220 ff. again summarizes the “reasons for the development of alternative gaits”.
Line 1226 should be “hard pace” to connet it with line 511 ff.
Line 1234 and 1308 attribute high lifting of the frontlimnbs to avoiding collisions with uneven ground. 1st: a similar conclusion was drawn earlier by Heckner-Bisping & Gräff, 1992 and Heckner-Bisping & Geier, 1993, both in Z.Säugetierkunde; 2nd lift of the hooves has to do with shorting of the leg pendulum during the foreswing, and so making the latter faster (Witte et al., 1995: Die Gangarten der Pferde., Teile I und II, Pferdeheilkunde) .
Lines 1247, 1252, 1254, 1263, 1266, 1269, 1287(!),: instead of pace better “hard pace” to maintain the same word for the same thing (“Discours sur la méthode”, R. Descartes in the 17th century).
Line 1350: “harness racers” is only true if pacing is counted among the “alternative gaits”, not as a variant of the trot (see my note above).
Lines 1389 – 1390: The last sentence seems to be less fluent than the earlier text. It may be superfluous, since it is more a declaration of the author´s opinion than a fact.
Author Response
Thank you so much for the very detailed review. Here are my responses to your very helpful comments.
Point 1. The article is intended to give a comprehensive overview on the gaits of horses. This aim is reached, although the greatest stress is placed on the “alternative gaits”. Working since years on horses, the reviewer congratulates the author for his impressive compilation which includes as many as 405 citations. In spite of the great number of my remarks (distributed on nearly 30 pages!), all of them are “minor corrections”. As soon as these are made, I strongly recommend to accept and publish this paper.
Response 1. Thanks so much. Very appreciated.
Point 2. Abstract, line 1: “Origination” should be replaced by the shorter and clearer “origin”
Response 2. So changed.
Point 3. Introduction, Lines 44 and 45: “...gallop (or canter)...” looks like addition of a synonym. This is o.k., but on p. 2, 1st line, the reader finds: “..the four-beat gallop (or three-beat canter) is a ...fast gait....”. This is confusing, because the so-called four-beat gallop occurs only at slow speeds, sometimes intended by the rider, sometimes not. Three beats are typical for the fastest among gaits, only at very high speeds (>15m/s, “Renngalopp” in German, in English “gallop” in contrast to the slower, three-beat “canter”) just 2 double beats can be heard, which can be interpreted as “four beats”.
Response 3. This is very helpful. Gallop and canter are distinguished (by beat) and more detail provided on number of beats. It is tricky as the terms canter and gallop often overlap in meaning.
Point 4. Lines 60, 62, 64 “...four limb suspension” is difficult to understand. “Suspension” means without ground contact. This is clear, but why the addition of “four limbs”.
Response 4. I clarified this by noting this means suspended phases when all four legs are off the ground at once; some authors also talk about suspended phases when both front limbs or both hind limbs are off the ground at the same time.
Point 5. 92 ff, 106 ff: The taxonomy of wild equids in this article is unusual, since Asian wild asses are lumped with African wild asses as E. (asinus), sometimes E. hemionus. All wild-living equids are confined to using the classical gaits: walk, trot, gallop.
Response 5. Thank you I made sure to distinguish the Asian and African wild asses as hemiones most properly refers to (some) of the Asiatic species.
Point 6. Line 108; “.. gallop (and sometimes canter)...” implies a difference between the two gaits, which does not exist. If the author just deletes “and sometimes”. this problem is solved. I personally prefer to calling “canter” the three-beat locomotion which can be maintained over long distances, and “gallop” the four-beat locomotion used in racing over short distances of 2400 m or less (see above, the English meaning)
Response 6. Thanks I clarified and distinguished the canter as 3-beat and gallop as 4-beat (or sometimes 2-beat).
Point 7. Lines 133-134 I do not know anything about a “park trot” of Morgan horses, and the sentence may therefore stay as before. But I am definitely missing the “passage”, a sustained trot, which is performed by all dressage horses.
Response 7. Passage gait added.
Point 8. Lines 140-141 This can be understood as a precise description of the “gallop” – as noted above, on lines 44 f. I just do not comprehend, why this is summarised under the headline “alternative lateral horse gaits.”
Response 8. All ‘natural’ gaits including gallop moved out of this section and into introduction.
Point 9. Fig. 1: The caption should explain what the numbers at the far ends of the rays do mean.
Response 9. Abbreviations and percentage explanations added to caption for figure 1.
Point 10. Lines 180 ff. The author indulges in mechanical ideas concerning the propulsion of the horses. This must be taken with care, because no really convincing evidence is presented.
Response 10. A couple of references are cited but yes this is an area for future research.
Point 11. Line 187 f.: The “rear limbs will have a low, sweeping stride”, but the “front limbs a short elevated stride...” This seems to ignore the simple fact, that stride length in both limbs must necessarily be the same (unless the animal becomes longer and longer or shorter and shorter with each step.
Response 11. Yes, this is not worded clearly, reworded so it is clear stride lengths are same but elevation of hoofs or angle of joints often different between front and rear limbs.
Point 12. Line 192: I am not entirely happy with the repeated note that striding follows the laws of the “inverted pendulum”. This is true for the stance phases, but self-evident for any locomotion on limbs. The swing phases follow the laws of a suspended, hanging pendulum, and this explains in many cases why the limbs are elevated (to shorten the limb pendulum and therefore to increase the speed of the foreswing), not in order to avoid obstacles – as the author says repeatedly.
Response 11. Interesting point; I have tried to follow authors here, references now added, who contrast mechanics of horse gaits based on those that ‘mainly’ follow an inverted pendulum (walk, running walk) vs. spring mass (rack, trot, pace) vs. collisional forces (gallop). Also interesting alternative hypothesis for elevation of front hooves. I now present both views.
Point 12. Line 220: “Heels” should be placed in quotation marks, to emphasise that the “heels of the hove” are mentioned in the meaning of blacksmiths.
Response 12. So done.
Point 13. Line 510 ff. – 583 deals with “hard pace, or flying pace”, a gait I would consider (in another context, however) together with trot.
Response 13. Yes similar gaits in terms of two beats; here I separate them as hard pace included with other alternative lateral gaits (due to uncommon nature and footfalls). See comment below also.
Point 14. Line 565 – 759 are the most elaborate recordings of historical literature which is known to me! I can only repeat what he says. The citations are more exact and detailed than anything I have seen before!
Response 14. Thanks I spent a lot of time researching this.
Point 15. Line 864: Wouldn´t be “grassy” better than “glassy”?
Response 15. Thanks, yes a typo; corrected.
Point 16. Lines 908 – 921 contain estimated absolute ages. These assumptions in my opinion are risky and perhaps not correct. They do not become more reliable if repeated on lines 948 and 949.
Response 16. Yes, I am following age dates given in cited references, usually based on dated ash bed close in time to prints. I consider these as the best temporal dates given what we know.
Point 17. Lines 956-974 review early domestication of horses, and consider the allele A of DTRM3 gene, which seems to characterise alternative gaits since nearly 10.000 years ago. An interesting, and not usually recognized item is that Philip of Macedonia imported in the year 339 BC Ferghana horses, earlier than the famous Chinese expeditions. In addition the Scythian culture in the east, as well as the Celts in the west are connected with gaited horses. What the author says, deviates from common views, but sound very plausible.
Response 17. Thank you. I have now added possible link of Celts to Scythians as well.
Point 18. Understanding of the author is not improved by his using uncommon abbreviations (CE, BCE)
Response 18. I am perfectly fine with BC and AD instead of CE and BCE and have so modified the abbreviations for dates; from what I can tell journal allows either set of abbreviations for dates but editors can make final decision.
Point 19. Lines 1072 – 1087 draw a connection between the alternative gaits and the terrain on which the animals live. This part is repeated below in Chapter 6, and I do not see a reason why this aspect is dealt with at two places.
Response 19. Sections are combined now.
Point 20. Lines 1088 – 1217 go into detail concerning the frequency of the A-allele of DTRM3-gene –as well as other genes! - in breeds of gaited horses. I have not controlled, whether there is any repetition or contradiction to the above-noted (on lines 956-974).
Response 20. Thanks. No changes necessary.
Point 21. Chapter 6, lines 1220 ff. again summarizes the “reasons for the development of alternative gaits”.
Response 21. Earlier sections combined with these.
Point 22. Line 1226 should be “hard pace” to connect it with line 511 ff. Lines 1247, 1252, 1254, 1263, 1266, 1269, 1287(!),: instead of pace better “hard pace” to maintain the same word for the same thing (“Discours sur la méthode”, R. Descartes in the 17th century).
Response 22. Great point; I know Descartes well; so modified.
Point 23. Lines 1234 and 1308 attribute high lifting of the front limbs to avoiding collisions with uneven ground. 1st: a similar conclusion was drawn earlier by Heckner-Bisping & Gräff, 1992 and Heckner-Bisping & Geier, 1993, both in Z. Säugetierkunde; 2nd lift of the hooves has to do with shorting of the leg pendulum during the foreswing, and so making the latter faster (Witte et al., 1995: Die Gangarten der Pferde., Teile I und II, Pferdeheilkunde).
Response 23. Thank you, these references and both hypotheses now presented in more detail.
Point 24. Line 1350: “harness racers” is only true if pacing is counted among the “alternative gaits”, not as a variant of the trot (see my note above).
Response 24. I do consider it among alternative lateral gaits as it is uncommon in many horse breeds (but yes similar to trot in other ways).
Point 25. Lines 1389 – 1390: The last sentence seems to be less fluent than the earlier text. It may be superfluous, since it is more a declaration of the author´s opinion than a fact.
Response 25. Added reference and further qualified very last sentence showing it is opinion of author.
Round 2
Reviewer 2 Report
Dear Author,
Thank you for the answers one by one and your objectivity, congratulations on the changes.
Its alterations were notorious, with the exception of one point, I consider the conclusion still excessive, still with references (should not happen), I suggest its reduction.
The other points have been changed.
Compliments, Reviewer.
Author Response
Point 1: With the exception of one point, I consider the conclusion still excessive, still with references (should not happen), I suggest its reduction.
Response 1: Thank you very much. I have altered the conclusion so it is reduced in length and also contains no references or dates. I have also made a couple of additional cuts in the length of the text as a whole and added one fact re Celtic war chariots.
Alan